# TimeBridge: Non-Stationarity Matters for Long-term Time Series Forecasting

## Abstract

Non-stationarity poses significant challenges for multivariate time series forecasting due to the inherent short-term fluctuations and long-term trends that can lead to spurious regressions or obscure essential long-term relationships. Most existing methods either eliminate or retain non-stationarity without adequately addressing its distinct impacts on short-term and long-term modeling. Eliminating non-stationarity is essential for avoiding spurious regressions and capturing local dependencies in short-term modeling, while preserving it is crucial for revealing long-term cointegration across variates. In this paper, we propose TimeBridge, a novel framework designed to *bridge the gap between non-stationarity and dependency modeling in long-term time series forecasting*. By segmenting input series into smaller patches, TimeBridge applies Integrated Attention to mitigate short-term non-stationarity and capture stable dependencies within each variate, while Cointegrated Attention preserves non-stationarity to model long-term cointegration across variates. Extensive experiments show that TimeBridge consistently achieves state-of-the-art performance in both short-term and long-term forecasting. Additionally, TimeBridge demonstrates exceptional performance in financial forecasting on the CSI 500 and S&P 500 indices, further validating its robustness and effectiveness. The code is available in the supplementary material.

## 1 Introduction

Multivariate time series forecasting aims to predict future changes based on historical observations of time series data, which holds significant applications in fields such as financial investment (Sezer et al., 2020), weather forecasting (Karevan & Suykens, 2020), and traffic flow prediction (Shu et al., 2021). However, the inherent non-stationarity of time series (Kim et al., 2022), characterized by short-term fluctuations and long-term trends, introduces challenges such as spurious regressions, making time series forecasting a particularly complex task.

Recently, many methods have emerged to utilize a normalization-and-denormalization paradigm to address non-stationarity in time series (Kim et al., 2022; Fan et al., 2023; Liu et al., 2023; 2024b). For instance, RevIN (Kim et al., 2022) normalizes the input data and subsequently applies its distributional characteristics to denormalize the output predictions. Building on this approach, other methods have designed distributional prediction networks (Fan et al., 2023) and more refined normalization techniques (Liu et al., 2023) to further mitigate non-stationarity. On the other hand, some studies (Liu et al., 2022b; Ma et al., 2024; Fan et al., 2024) argue that over-stabilizing time series may actually reduce the richness of embedded features, leading to a decline in model performance. Existing methods for addressing non-stationarity in time series face a dilemma: some prioritize eliminating non-stationary factors to reduce overfitting, while others attempt to incorporate these factors but lack comprehensive theoretical frameworks. Furthermore, they do not adequately explain the trade-offs between removing non-stationarity and leveraging it for modeling.

It is notable that non-stationary characteristics have distinct impacts on modeling short-term and long-term dependencies. Non-stationarity can lead to spurious regressions in short-term modeling due to the high randomness and unpredictability of short-term fluctuations (Noriega & Ventosa-Santaulària, 2007) (see Fig. 1b). For example, a sudden drop in temperature could be caused by a typhoon or cold front, both of which have no intrinsic connection. Retaining non-stationarity can result in false correlations between such unrelated events. However, non-stationarity is crucial

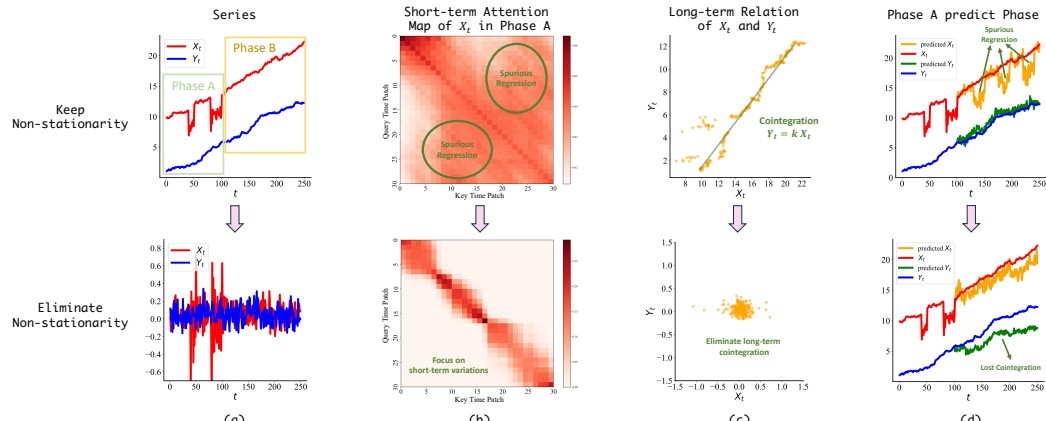

Figure 1: Visualization of the impact of non-stationarity on short-term and long-term modeling. (a) Comparison of two cointegrated series $X_t$ and $Y_t$ before and after de-trending, with $X_t$ showing two random fluctuations in Phase A. (b) Short-term patch attention map of $X_t$ in Phase A, showing that non-stationarity leads to spurious regression in short-term modeling. (c) Long-term relationship between $X_t$ and $Y_t$, where removing non-stationarity eliminates the cointegration. (d) Non-stationarity enables modeling long-term cointegration between variates but causes spurious regressions in short-term modeling (orange line), while de-trending benefits short-term modeling but disrupts long-term relationships (green line).

for capturing long-term cointegration relationships among variates, reflecting their co-movement or synchronized changes over time (Fanchon & Wendel, 1992). Removing non-stationarity may also eliminate these essential long-term dependencies (see Fig. 1c). Therefore, while non-stationarity can cause spurious regressions in short-term modeling, it is essential for modeling long-term dependencies between variates. Conversely, eliminating non-stationarity benefits short-term modeling but erases long-term cointegration (see Fig. 1d).

In this paper, to address the dual challenges posed by non-stationarity in short-term and long-term modeling, we propose distinct strategies tailored to each scenario. For short-term modeling, we eliminate non-stationarity to capture the strong temporal dependencies within each variate, as short-term causal relationships exist mainly between consecutive time points within a single variate rather than across variates. This strategy reduces the risk of spurious regressions from non-stationary fluctuations, enabling the model to better capture the local causal dynamics. For long-term modeling, we utilize the preserved non-stationarity to uncover long-term cointegration relationships between different variates, thereby enabling more accurate and reliable long-term forecasting.

Technically, based on the above motivations, we propose TimeBridge as a novel framework to *bridge the gap between non-stationarity and dependency modeling in long-term time series forecasting*. TimeBridge first captures short-term fluctuations by partitioning the input sequence into small-length patches, followed by utilizing Integrated Attention to model these stabilized sub-sequences within each variate. Here, "Integrated" reflects the non-stationary nature of the short-term series, also referred to as integrated series (Park & Phillips, 2001). Subsequently, we downsample the patches to reduce their quantity, thereby enriching each patch with more long-term information. Cointegrated Attention retains the non-stationary characteristics of the sequences to effectively capture the long-term cointegration relationships among variates. Experiments across multiple datasets demonstrate that TimeBridge achieves consistent state-of-the-art performance in both long-term and short-term forecasting. Furthermore, we validate the effectiveness of TimeBridge on two financial datasets, the CSI 500 and S&P 500, which exhibit significant short-term volatility and strong long-term cointegration relationships among sectors.

In a nutshell, our contributions are summarized in three folds:

1. Going beyond previous methods, we establish a novel connection between non-stationarity and dependency modeling, highlighting the importance of eliminating non-stationarity in short-term variations while preserving it for long-term cointegration.

2. We propose TimeBridge, a novel framework that employs Integrated Attention to model temporal dependencies by mitigating short-term non-stationarity, and Cointegrated Attention to capture long-term cointegration across variates while retaining non-stationarity.

3. Comprehensive experiments demonstrate that TimeBridge achieves state-of-the-art performance in both long-term and short-term forecasting across various datasets. Moreover, we further validate its robustness and effectiveness on the CSI 500 and S&P 500 indices, which pose additional challenges due to their complex volatility and cointegration characteristics.

## 2 RELATED WORKS

As shown in Fig. 2, recent advancements in multivariate time series forecasting have predominantly focused on two core directions: **Normalization** and **Dependecy Modeling**.

**Normalization** can be divided into stationary and non-stationary methods. Stationary methods (Kim et al., 2022; Fan et al., 2023; Liu et al., 2024b; 2023) aim to eliminate non-stationarity through model-agnostic normalization techniques, thereby preventing spurious regressions and enhancing model performance. For example, RevIN (Kim et al., 2022) applies Z-normalization to the input sequence and then reverses the normalization on the output using the distributional characteristics of the input, assuming that both share similar distributional properties. Dish-TS (Fan et al., 2023) takes this further by predicting the statistical characteristics of the output with a distribution prediction model. Additionally, SAN (Liu et al., 2023) offers a more granular patch-level prediction method. Conversely, some approaches (Liu et al., 2022b; Ma et al., 2024; Fan et al., 2024) advocate preserving non-stationarity, as excessive normalization can eliminate diverse sequence characteristics and limit predictive accuracy.

**Dependency Modeling** focuses on designing methods to capture the relationships within multivariate time series, which can be classified into Channel Independent (CI) and Channel Dependent (CD) methods. CI methods (Zeng et al., 2023; Das et al., 2023; Nie et al., 2023; Dai et al., 2024; Lin et al., 2024) rely exclusively on the historical values of each individual channel for prediction, deliberately avoiding cross-channel interactions. This strategy not only stabilizes the training process but also excels at capturing rapid temporal dynamics unique to each variate. In contrast, CD methods (Wu et al., 2021; Zhou et al., 2022; Wu et al., 2023; Zhang & Yan, 2023; Liu et al., 2024a) leverage the interrelationships between variates for modeling. While these methods utilize more information, they struggle with spurious regressions when modeling short-term dependencies, failing to capture rapid changes effectively.

The challenge with previous methods lies in their isolated treatment of non-stationarity and dependency modeling, overlooking their intrinsic connection. Due to non-stationarity, time series often exhibit significant short-term fluctuations, leading to severe spurious regressions when modeling short-term dependencies. However, capturing long-term cointegration requires preserving this underlying variability. Therefore, short-term random fluctuations need to be addressed by elimi-

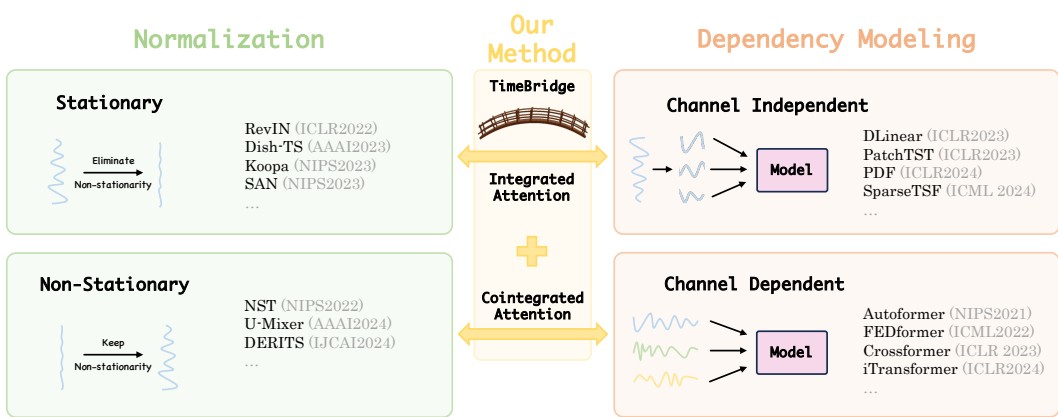

Figure 2: Time series forecasting methods categorized by normalization and dependency modeling.

nating non-stationarity and modeling intra-variate temporal dependencies, while long-term cointegration demands preserving non-stationarity for inter-variate modeling. Our proposed TimeBridge addresses these issues by employing Integrated Attention and Cointegrated Attention, respectively.

## 3 METHOD

In the task of multivariate time series forecasting, the objective is to predict future sequences $\mathbf{Y} = [\mathbf{x}_{I+1}, \cdots, \mathbf{x}_{I+O}] \in \mathbb{R}^{C \times O}$ given historical input sequences $\mathbf{X} = [\mathbf{x}_1, \cdots, \mathbf{x}_I] \in \mathbb{R}^{C \times I}$. Here, $I$ and $O$ denote the lengths of the input and output sequences, respectively, and $C$ represents the number of time variates. It is important to recognize that real-world time series data often exhibit high short-term uncertainty, while long-term dynamics may reveal cointegration relationships among different time variates.

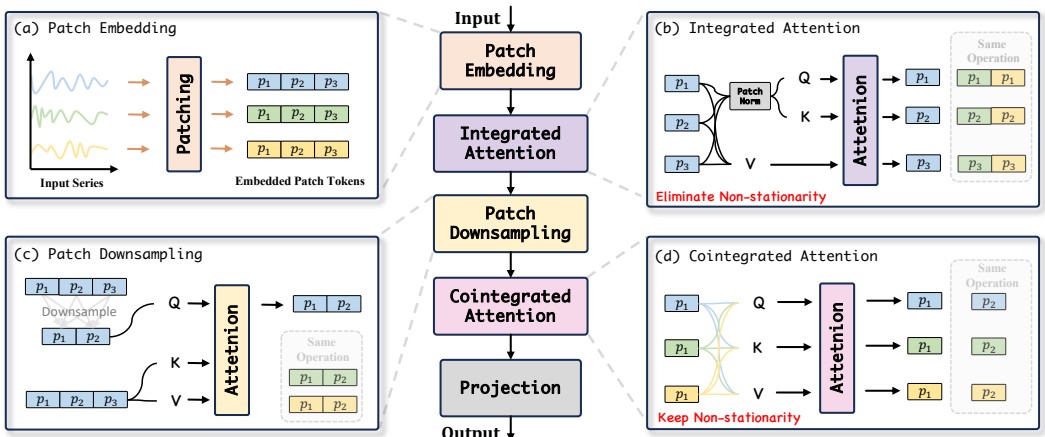

Figure 3: Overall architecture of TimeBridge: (a) **Patch Embedding** divides the input sequence into non-overlapping patches and embeds each as a token; (b) **Integrated Attention** models temporal dependencies within each variate by mitigating short-term non-stationarity; (c) **Patch Downsampling** reduce patches to aggregates long-term information and lower complexity; (d) **Cointegrated Attention** captures long-term relationships across variates while keeping non-stationarity.

### 3.1 STRUCTURE OVERVIEW

As illustrated in Fig. 3, our proposed TimeBridge consists of four key components: (a) **Patch Embedding** segments the input sequence into non-overlapping patches and transforms each patch into a patch token; (b) **Integrated Attention** models the dependencies among all patch tokens of the same variates. By eliminating non-stationarity within each patch token, it mitigates the risk of spurious regressions that could arise from abrupt short-term changes; (c) **Patch Downsampling** aggregates global information and reduces the number of patches to encapsulate richer long-term features within each patch, while simultaneously lowering computational complexity; (d) **Cointegrated Attention** preserves the non-stationary characteristics of the sequence and models the long-term cointegration relationships across different variates within the same temporal window.

### 3.2 PATCH EMBEDDING

In this stage, each variate of the input sequence $\mathbf{X}$ is first divided into non-overlapping patches, and each patch is then mapped to an embedded patch token. Since the process is identical for each variate, we use $\mathbf{X}$ to represent a single variate and later restore the dimensionality of the variates. Formally, this process can formulated as follows:

$$\{\mathbf{p}_1, \cdots, \mathbf{p}_N\} = \text{Patching}(\mathbf{X}), \quad \mathbf{P} = \text{Embedding}(\mathbf{p}_1, \cdots, \mathbf{p}_N) \tag{1}$$

Here, each patch $\mathbf{p}_i$ has a length of $S$, and the number of patches $N = \lfloor \frac{I}{S} \rfloor$. The Embedding($\cdot$) operation transforms each patch from its original length $S$ to a hidden dimension $D$ through a trainable linear layer. This results in embedded patch tokens $\mathbf{P} \in \mathbb{R}^{C \times N \times D}$, where each of the $C$ variates contains $N$ patches, capturing local information that is typically subject to rapid short-term fluctuations. For convenience, we denote $\mathbf{P}_{c,:}$ as the set of all patches within a single variate and $\mathbf{P}_{:,n}$ as the patches across all variates at the same time position in the following sections.

### 3.3 INTEGRATED ATTENTION

The embedded patch tokens $\mathbf{P}$ represent short-term non-stationary sequences, also referred to as integrated series of order $k$ ($k > 0$) (Park & Phillips, 2001; Mushtaq, 2011). This non-stationarity makes it challenging to model dependencies across different variates, as short-term fluctuations are highly susceptible to external shocks. Furthermore, modeling temporal dependencies within the same variate can lead to spurious regression due to the inherent non-stationarity of the patches. To address this, we first apply a patch-wise normalization to all patches within a variate:

$$\mathbf{p}_i^{\text{Trend}} = \text{AvgPool}(\text{Padding}(\mathbf{p}_i)), \quad \mathbf{p}_i' = \mathbf{p}_i - \mathbf{p}_i^{\text{Trend}}, \quad \mathbf{P}_{c,:}' = \{\mathbf{p}_1', \cdots, \mathbf{p}_N'\}, \tag{2}$$

where the AvgPool($\cdot$) operation is moving average with the Padding($\cdot$) operation to keep the series length unchanged. We then employ the proposed Integrated Attention mechanism to capture temporal dependencies within the same variate:

$$\hat{\mathbf{P}}_{c,:} = \text{LayerNorm}\left(\mathbf{P}_{c,:} + \text{Attention}(\mathbf{P}_{c,:}', \mathbf{P}_{c,:}', \mathbf{P}_{c,:})\right), \tag{3}$$

$$\mathbf{P}_{c,:} = \text{LayerNorm}\left(\hat{\mathbf{P}}_{c,:} + \text{MLP}(\hat{\mathbf{P}}_{c,:})\right), \tag{4}$$

where MLP($\cdot$) represents a multi-layer feedforward network, and LayerNorm($\cdot$) denotes layer normalization. The attention mechanism uses the normalized $\mathbf{P}_{c,:}'$ as both Query and Key, while the original $\mathbf{P}_{c,:}$ serves as the Value. This design generates a stationary attention map, which is then directly multiplied by the Value, removing the need for subsequent denormalization. By leveraging Integrated Attention in this way, we effectively model the temporal dependencies without being affected by the short-term non-stationary nature of the sequences.

### 3.4 PATCH DOWNSAMPLING

Long-term equilibrium relationships between sequences, or cointegration among different variates, often require sequences to contain sufficient long-term information to emerge. Therefore, before modeling the cointegration between variates, it is crucial to increase the amount of global information represented by each patch. This is achieved by reducing the number of patches and aggregating global information through the attention mechanism:

$$\mathbf{P}_{c,:}' = \text{Downsample}(\mathbf{P}_{c,:}), \quad \mathbf{P}_{c,:} = \text{Attention}(\mathbf{P}_{c,:}', \mathbf{P}_{c,:}, \mathbf{P}_{c,:}). \tag{5}$$

Here, Downsample($\cdot$) reduces the $N$ patches in $\mathbf{P}_{c,:}$ to $M$ patches ($M < N$) using an MLP. By employing the downsampled $\mathbf{P}_{c,:}' \in \mathbb{R}^{M \times D}$ as the Query and the original $\mathbf{P}_{c,:} \in \mathbb{R}^{N \times D}$ as the Key and Value in the attention mechanism, we leverage the long-range modeling capability of attention to dynamically aggregate global information. This allows each patch to encapsulate richer long-term information, making it possible to capture the intricate cointegration relationships that emerge only over sufficiently extended temporal horizons.

### 3.5 COINTEGRATED ATTENTION

Although short-term relationships between integrated series are susceptible to spurious regressions, accurately modeling long-term cointegration between sequences necessitates retaining their inherent non-stationary characteristics. Since each downsampled patch now encapsulates more extensive

long-term information, we leverage Cointegrated Attention to directly model the cointegration relationships among all variates at the same time interval $\mathbf{P}_{:,n} \in \mathbb{R}^{C \times D}$:

$$\hat{\mathbf{P}}_{:,n} = \text{LayerNorm}(\mathbf{P}_{:,n} + \text{Attention}(\mathbf{P}_{:,n}, \mathbf{P}_{:,n}, \mathbf{P}_{:,n})), \tag{6}$$

$$\mathbf{P}_{:,n} = \text{LayerNorm}(\hat{\mathbf{P}}_{:,n} + \text{MLP}(\hat{\mathbf{P}}_{:,n})). \tag{7}$$

This attention mechanism not only captures the global cointegration relationships across variates but also adaptively assesses the strength of these relationships: stronger cointegration is reflected by higher attention weights, while weaker connections receive lower weights. Finally, the embedded patch tokens $\mathbf{P} \in \mathbb{R}^{C \times M \times D}$ are unpatched and projected to the final output $\mathbf{Y} \in \mathbb{R}^{C \times O}$.

## 4 EXPERIMENT

To validate the effectiveness of the proposed TimeBridge, we conduct extensive experiments on a variety of time series forecasting tasks, including both long-term and short-term forecasting. Additionally, we evaluate TimeBridge on financial forecasting tasks characterized by significant short-term volatility and strong long-term cointegration relationships among sectors.

**Baselines.** For long-term forecasting, we select a diverse set of state-of-the-art baselines representative of recent advancements in time series forecasting, including the Transformer-based PDF (Dai et al., 2024), the CNN-based ModernTCN (Donghao & Xue, 2024), the MLP-based TimeMixer (Wang et al., 2024), as well as other competitive methods such as iTransformer (Liu et al., 2024a), PatchTST (Nie et al., 2023), Crossformer (Zhang & Yan, 2023), FEDformer (Zhou et al., 2022), MICN (Wang et al., 2022), TimesNet (Wu et al., 2023), and DLinear (Zeng et al., 2023). For short-term forecasting, we add a well-performing baseline SCINet (Liu et al., 2022a). For financial forecasting, we also incorporate the momentum strategy CSM (Jegadeesh & Titman, 1993) and the reversal strategy BLSW (Poterba & Summers, 1988), along with two classic deep learning models, LSTM (Hochreiter & Schmidhuber, 1997) and Transformer (Vaswani et al., 2017a), to provide a comprehensive evaluation.

**Implementation Details.** All experiments are implemented in PyTorch (Paszke et al., 2019) and conducted on two NVIDIA RTX 3090 24GB GPUs. We use the Adam optimizer (Kingma, 2014) with a learning rate selected from $\{1e\text{-}3, 1e\text{-}4, 5e\text{-}4\}$. The number of patches $N$ is set to 30. For additional details on hyperparameters and settings, please refer to the Appendix D.

### 4.1 LONG-TERM FORECASTING

**Setups.** We conduct long-term forecasting experiments on eight widely-used real-world datasets, including the Electricity Transformer Temperature (ETT) dataset with its four subsets (ETTh1, ETTh2, ETTm1, ETTm2), as well as Weather, Electricity, Traffic, and Solar (Liu et al., 2024a). These datasets exhibit strong non-stationary characteristics, detailed in Appendix C. Following previous works (Zhou et al., 2021; Wu et al., 2021), we use Mean Square Error (MSE) and Mean Absolute Error (MAE) as evaluation metrics. We set the input length $I$ to 720 for our method. For other baselines, we adopt two settings: one uses the original results from the baseline papers, and the other involves searching for the optimal input length $I$ and other hyperparameters. Details of the search process can be found in Appendix E.1.

**Results.** As shown in Tab. 1, TimeBridge consistently outperforms other methods on these non-stationary datasets, with an average improvement of over 10% compared to the baselines. Moreover, in experiments with different hyperparameter search settings in Tab. 2, TimeBridge continues to achieve the best overall results. Specifically, compared to the current state-of-the-art methods—Transformer-based PDF (Dai et al., 2024), CNN-based ModernTCN (Donghao & Xue, 2024), and MLP-based TimeMixer (Wang et al., 2024)—TimeBridge reduces MSE and MAE by 3.10%/1.64%, 3.55%/0.81%, and 6.92%/4.54%, respectively.

| Models | TimeBridge (Ours) | | iTransformer (2024a) | | PDF (2024) | | TimeMixer (2024) | | PatchTST (2023) | | Crossformer (2023) | | FEDformer (2022) | | ModernTCN (2024) | | MICN (2022) | | TimesNet (2023) | | DLinear (2023) | |
|---|---|---|---|---|---|---|---|---|---|---|---|---|---|---|---|---|---|---|---|---|---|---|
| Metric | MSE | MAE | MSE | MAE | MSE | MAE | MSE | MAE | MSE | MAE | MSE | MAE | MSE | MAE | MSE | MAE | MSE | MAE | MSE | MAE | MSE | MAE |
| ETTm1 | 0.353 | 0.388 | 0.407 | 0.410 | **0.339** | **0.372** | 0.381 | 0.395 | 0.353 | 0.382 | 0.431 | 0.443 | 0.448 | 0.452 | 0.351 | 0.381 | 0.392 | 0.414 | 0.400 | 0.406 | 0.357 | 0.379 |
| ETTm2 | **0.246** | **0.310** | 0.288 | 0.332 | 0.252 | 0.313 | 0.275 | 0.323 | 0.256 | 0.317 | 0.621 | 0.510 | 0.305 | 0.349 | 0.253 | 0.314 | 0.290 | 0.343 | 0.291 | 0.333 | 0.267 | 0.332 |
| ETTh1 | **0.399** | 0.420 | 0.454 | 0.447 | **0.399** | 0.419 | 0.447 | 0.440 | 0.413 | 0.434 | 0.446 | 0.464 | 0.440 | 0.460 | 0.404 | 0.420 | 0.440 | 0.462 | 0.458 | 0.450 | 0.423 | 0.437 |
| ETTh2 | 0.346 | 0.394 | 0.383 | 0.407 | 0.327 | **0.376** | 0.364 | 0.395 | 0.324 | 0.381 | 0.835 | 0.675 | 0.434 | 0.447 | **0.322** | 0.379 | 0.411 | 0.440 | 0.414 | 0.427 | 0.431 | 0.447 |
| Weather | **0.218** | **0.259** | 0.258 | 0.279 | 0.225 | 0.261 | 0.240 | 0.271 | 0.226 | 0.264 | 0.343 | 0.386 | 0.309 | 0.360 | 0.224 | 0.264 | 0.243 | 0.299 | 0.259 | 0.287 | 0.246 | 0.300 |
| Electricity | **0.149** | **0.246** | 0.178 | 0.270 | 0.159 | 0.250 | 0.182 | 0.272 | 0.159 | 0.253 | 0.293 | 0.351 | 0.205 | 0.315 | 0.156 | 0.253 | 0.187 | 0.295 | 0.192 | 0.295 | 0.166 | 0.264 |
| Traffic | **0.360** | **0.255** | 0.428 | 0.282 | 0.383 | 0.254 | 0.484 | 0.297 | 0.391 | 0.264 | 0.535 | 0.300 | 0.573 | 0.347 | 0.396 | 0.270 | 0.542 | 0.316 | 0.620 | 0.336 | 0.434 | 0.295 |
| Solar† | **0.181** | **0.239** | 0.233 | 0.262 | 0.205 | 0.265 | 0.216 | 0.280 | 0.207 | 0.294 | 0.204 | 0.248 | 0.296 | 0.407 | 0.228 | 0.282 | 0.247 | 0.296 | 0.319 | 0.348 | 0.329 | 0.400 |

† For papers that do not originally report results on the Solar dataset, we run experiments using their official code with default settings.

Table 1: Long-term forecasting results from the original papers. All results are averaged across four different prediction lengths: $O \in \{96, 192, 336, 720\}$. The best and second-best results are highlighted in **bold** and underlined, respectively. See Tab. 10 for full results.

| Models | TimeBridge (Ours) | | iTransformer (2024a) | | PDF (2024) | | TimeMixer (2024) | | PatchTST (2023) | | Crossformer (2023) | | FEDformer (2022) | | ModernTCN (2024) | | MICN (2022) | | TimesNet (2023) | | DLinear (2023) | |
|---|---|---|---|---|---|---|---|---|---|---|---|---|---|---|---|---|---|---|---|---|---|---|
| Metric | MSE | MAE | MSE | MAE | MSE | MAE | MSE | MAE | MSE | MAE | MSE | MAE | MSE | MAE | MSE | MAE | MSE | MAE | MSE | MAE | MSE | MAE |
| ETTm1 | 0.353 | 0.388 | 0.362 | 0.391 | **0.335** | **0.373** | 0.348 | 0.375 | 0.353 | 0.382 | 0.420 | 0.435 | 0.382 | 0.422 | 0.351 | 0.381 | 0.383 | 0.406 | 0.400 | 0.406 | 0.357 | 0.379 |
| ETTm2 | **0.246** | **0.310** | 0.269 | 0.329 | 0.247 | 0.311 | 0.256 | 0.315 | 0.256 | 0.317 | 0.518 | 0.501 | 0.292 | 0.343 | 0.253 | 0.314 | 0.277 | 0.336 | 0.291 | 0.333 | 0.267 | 0.332 |
| ETTh1 | 0.399 | 0.420 | 0.439 | 0.448 | **0.395** | **0.420** | 0.411 | 0.423 | 0.413 | 0.434 | 0.440 | 0.463 | 0.428 | 0.454 | 0.404 | **0.420** | 0.433 | 0.462 | 0.458 | 0.450 | 0.423 | 0.437 |
| ETTh2 | 0.346 | 0.394 | 0.374 | 0.406 | 0.326 | **0.376** | 0.316 | 0.384 | 0.324 | 0.381 | 0.809 | 0.658 | 0.388 | 0.434 | 0.322 | 0.379 | 0.385 | 0.430 | 0.414 | 0.427 | 0.431 | 0.447 |
| Weather | **0.218** | **0.259** | 0.233 | 0.271 | 0.220 | 0.259 | 0.222 | 0.262 | 0.226 | 0.264 | 0.228 | 0.287 | 0.305 | 0.287 | 0.224 | 0.264 | 0.242 | 0.298 | 0.259 | 0.287 | 0.240 | 0.300 |
| Electricity | **0.149** | **0.246** | 0.164 | 0.261 | 0.156 | 0.250 | 0.156 | **0.246** | 0.159 | 0.253 | 0.181 | 0.279 | 0.205 | 0.315 | 0.156 | 0.253 | 0.182 | 0.292 | 0.192 | 0.295 | 0.166 | 0.264 |
| Traffic | **0.360** | **0.255** | 0.397 | 0.282 | 0.377 | 0.256 | 0.387 | 0.262 | 0.391 | 0.264 | 0.523 | 0.284 | 0.573 | 0.347 | 0.396 | 0.270 | 0.535 | 0.312 | 0.620 | 0.336 | 0.434 | 0.295 |
| Solar | **0.181** | **0.239** | 0.200 | 0.260 | 0.205 | 0.265 | 0.192 | 0.244 | 0.194 | 0.245 | 0.191 | 0.242 | 0.243 | 0.350 | 0.228 | 0.282 | 0.213 | 0.266 | 0.244 | 0.334 | 0.247 | 0.309 |

Table 2: Long-term forecasting hyperparameter search results. All results are averaged across four different prediction lengths: $O \in \{96, 192, 336, 720\}$. See Tab. 11 for full results.

## 4.2 SHORT-TERM FORECASTING

**Setups.** For short-term forecasting, we conduct experiments on the PeMS datasets (Wang et al., 2024), which capture complex spatiotemporal correlations among multiple variates across city-wide traffic networks. We use mean absolute error (MAE), mean absolute percentage error (MAPE), and root mean squared error (RMSE) as evaluation metrics. The input length $I$ is set to 96 and the output length $O$ to 12 for all baselines. Details of datasets and metrics are in Appendix C and Appendix B.2.

**Results.** As shown in Tab. 3, methods that perform well in long-term forecasting with channel-independent approaches, such as PatchTST (Nie et al., 2023) and DLinear (Zeng et al., 2023), suffer from significant performance degradation on the PeMS dataset due to its strong inter-variable dependencies. In contrast, TimeBridge demonstrates robust performance on this challenging task, outperforming even the recent state-of-the-art method TimeMixer (Wang et al., 2024), which highlights its effectiveness in capturing complex spatiotemporal relationships.

| Models | | TimeBridge (Ours) | TimeMixer (2024) | SCINet (2022a) | Crossformer (2023) | PatchTST (2023) | TimesNet (2023) | MICN (2022) | DLinear (2023) | FEDformer (2022) | Stationary (2022b) | Autoformer (2021) | Informer (2021) |
|---|---|---|---|---|---|---|---|---|---|---|---|---|---|
| | MAE | **14.52** | 14.63 | 15.97 | 15.64 | 18.95 | 16.41 | 15.71 | 19.70 | 19.00 | 17.64 | 18.08 | 19.19 |
| PeMS03 | MAPE | **14.21** | 14.54 | 15.89 | 15.74 | 17.29 | 15.17 | 15.67 | 18.35 | 18.57 | 17.56 | 18.75 | 19.58 |
| | RMSE | **23.10** | 23.28 | 25.20 | 25.56 | 30.15 | 26.72 | 24.55 | 32.35 | 30.05 | 28.37 | 27.82 | 32.70 |
| | MAE | 19.24 | **19.21** | 20.35 | 20.38 | 24.86 | 21.63 | 21.62 | 24.62 | 26.51 | 22.34 | 25.00 | 22.05 |
| PeMS04 | MAPE | **12.42** | 12.53 | 12.84 | 12.84 | 16.65 | 13.15 | 13.53 | 16.12 | 16.76 | 14.85 | 16.70 | 14.88 |
| | RMSE | 31.12 | **30.92** | 32.31 | 32.41 | 40.46 | 34.90 | 34.39 | 39.51 | 41.81 | 35.47 | 38.02 | 36.20 |
| | MAE | **20.43** | 20.57 | 22.79 | 22.54 | 27.87 | 25.12 | 22.28 | 28.65 | 27.92 | 26.02 | 26.92 | 27.26 |
| PeMS07 | MAPE | **8.42** | 8.62 | 9.41 | 9.38 | 12.69 | 10.60 | 9.57 | 12.15 | 12.29 | 11.75 | 11.83 | 11.63 |
| | RMSE | **33.44** | 33.59 | 35.61 | 35.49 | 42.56 | 40.71 | 35.40 | 45.02 | 42.29 | 42.34 | 40.60 | 45.81 |
| | MAE | **14.98** | 15.22 | 17.38 | 17.56 | 20.35 | 19.01 | 17.76 | 20.26 | 20.56 | 19.29 | 20.47 | 20.96 |
| PeMS08 | MAPE | **9.56** | 9.67 | 10.80 | 10.92 | 13.15 | 11.83 | 10.76 | 12.09 | 12.41 | 12.21 | 12.27 | 13.20 |
| | RMSE | **23.77** | 24.26 | 27.34 | 27.21 | 31.04 | 30.65 | 27.26 | 32.38 | 32.97 | 38.62 | 31.52 | 30.61 |

Table 3: Short-term forecasting results in the PeMS datasets.

## 4.3 FINANCIAL FORECASTING

**Setups.** We conduct experiments on both the U.S. and Chinese stock markets, including the S&P 500 and CSI 500 indices. Stock price movements are influenced by various factors such as economic indicators, market sentiment, geopolitical events, and company-specific news, leading to high non-stationarity. We predict next-day returns using historical data and generate investment portfolios with a buy-hold-sell strategy (Sanderson & Lumpkin-Sowers, 2018). At day $t + 1$ open, traders sell day $t$ stocks and buy top-ranked ones based on predicted returns. Following previous work (Lin et al., 2021; Li et al., 2024), we evaluate performance using Annual Return Ratio (ARR), Annual Volatility (AVol), Maximum Drawdown (MDD), Annual Sharpe Ratio (ASR), Calmar Ratio (CR), and Information Ratio (IR). Details of datasets and metrics are in Appendix C and Appendix B.3.

**Results.** As shown in Tab. 4, due to the non-stationary dynamics and complex market dependencies, other baseline methods struggle to consistently and accurately identify broadly optimal portfolios across different markets. In contrast, TimeBridge consistently performs best in both financial markets, demonstrating its ability to capture short-term fluctuations within financial time series and long-term cointegration between sectors.

| Models | CSI 500 | | | | | | S&P 500 | | | | | |
|---|---|---|---|---|---|---|---|---|---|---|---|---|
| | ARR↑ | AVol↓ | MDD↓ | ASR↑ | CR↑ | IR↑ | ARR↑ | AVol↓ | MDD↓ | ASR↑ | CR↑ | IR↑ |
| BLSW (1988) | 0.110 | 0.227 | -0.155 | 0.485 | 0.710 | 0.446 | 0.199 | 0.318 | -0.223 | 0.626 | 0.892 | 0.774 |
| CSM (1993) | 0.015 | 0.229 | -0.179 | 0.066 | 0.084 | 0.001 | 0.099 | 0.250 | -0.139 | 0.396 | 0.712 | 0.584 |
| LSTM (1997) | -0.008 | 0.159 | -0.172 | -0.047 | -0.044 | -0.128 | 0.142 | 0.162 | -0.178 | 0.877 | 0.798 | 0.929 |
| Transformer (2017a) | 0.154 | 0.156 | -0.135 | 0.986 | 1.143 | 0.867 | 0.135 | 0.159 | -0.140 | 0.852 | 0.968 | 0.908 |
| PatchTST (2023) | 0.118 | **0.152** | -0.127 | 0.776 | 0.923 | 0.735 | 0.146 | 0.167 | -0.140 | 0.877 | 1.042 | 0.949 |
| Crossformer (2023) | -0.039 | 0.163 | -0.217 | -0.238 | -0.179 | -0.350 | 0.284 | **0.159** | **-0.114** | 1.786 | **2.491** | 1.646 |
| iTransformer (2024a) | 0.214 | 0.168 | -0.164 | 1.276 | 1.309 | 1.173 | 0.159 | 0.170 | -0.139 | 0.941 | 1.150 | 0.955 |
| TimeMixer (2024) | 0.078 | 0.153 | **-0.114** | 0.511 | 0.685 | 0.385 | 0.254 | 0.162 | -0.131 | 1.568 | 1.938 | 1.448 |
| TimeBridge (**Ours**) | **0.285** | 0.203 | -0.196 | **1.405** | **1.453** | **1.317** | **0.326** | 0.169 | -0.142 | **1.927** | 2.298 | **1.842** |

Table 4: Results for financial time series forecasting in CSI 500 and S&P 500 datasets. See Tab. 12 for full results.

## 5 ABLATION STUDIES

To validate the effectiveness of the proposed TimeBridge, we conduct a comprehensive ablation study on its architectural design. In Tab. 5, Tab. 6, and Tab. 7, the rows highlighted in `gray` correspond to the original TimeBridge configuration, serving as a baseline for comparison with various modified versions.

**Ablation on removing or keeping non-stationarity.** We conduct the following experiments: ① Non-stationarity retained in both Integrated and Cointegrated Attention. ② Retained in Integrated, removed from Cointegrated. ③ Removed from Integrated, retained in Cointegrated. ④ Removed from both. Results in Tab. 5 show that the best performance is achieved when non-stationarity is removed in Integrated Attention, which models short-term intra-variate fluctuations, and retained in Cointegrated Attention, which captures long-term inter-variate dependencies. Conversely, retaining non-stationarity in Integrated Attention while removing it from Cointegrated Attention yields the worst results.

| Case | Integrated Attention + Norm? | Cointegrated Attention + Norm? | Weather | | Solar | | Electricity | | Traffic | |
|---|---|---|---|---|---|---|---|---|---|---|
| | | | MSE | MAE | MSE | MAE | MSE | MAE | MSE | MAE |
| ① | × | × | 0.220 | 0.260 | 0.183 | 0.242 | 0.153 | 0.249 | 0.371 | 0.260 |
| ② | × | ✓ | 0.220 | 0.260 | 0.183 | 0.252 | 0.155 | 0.251 | 0.381 | 0.263 |
| ③ | ✓ | × | **0.218** | **0.259** | **0.181** | **0.239** | **0.149** | **0.246** | **0.360** | **0.255** |
| ④ | ✓ | ✓ | 0.219 | **0.259** | 0.183 | 0.242 | 0.153 | 0.250 | 0.374 | 0.289 |

Table 5: Ablation on the effect of removing non-stationarity in Integrated Attention and Cointegrated Attention. ✓ indicates the use of patch normalization to eliminate non-stationarity, while × means non-stationarity is retained.

**Ablation on Integrated and Cointegrated Attention impact and order.** We conduct the following experiments: ① Integrated Attention only, ② Cointegrated Attention only, ③ Integrated Attention followed by Cointegrated Attention, and ④ Cointegrated Attention followed by Integrated Attention, with patch downsampling replaced by upsampling in this case. The results in Tab. 6 show that both ① and ② underperform compared to ③, indicating that both components are beneficial. Additionally, ④ shows the weakest performance, possibly because modeling long-term cointegrated relationships first leads to a loss of important short-term temporal features.

| Case | Integrated Attention | Cointegrated Attention | Weather | | Solar | | Electricity | | Traffic | |
|---|---|---|---|---|---|---|---|---|---|---|
| | Order | Order | MSE | MAE | MSE | MAE | MSE | MAE | MSE | MAE |
| ① | 1 | × | 0.220 | 0.262 | 0.184 | 0.244 | 0.158 | 0.252 | 0.388 | 0.264 |
| ② | × | 1 | 0.222 | 0.264 | 0.191 | 0.260 | 0.165 | 0.263 | 0.369 | 0.265 |
| ③ | 1 | 2 | **0.218** | **0.259** | **0.181** | **0.239** | **0.149** | **0.246** | **0.360** | **0.255** |
| ④ | 2 | 1 | 0.227 | 0.266 | 0.190 | 0.252 | 0.160 | 0.255 | 0.396 | 0.281 |

Table 6: Ablation on the impact and order of Integrated Attention and Cointegrated Attention. "Order" specifies the sequence, with lower numbers indicating earlier placement. × indicates the component is removed.

**Ablation on modeling approaches for Integrated Attention and Cointegrated Attention.** We conduct the following experiments: ① both Integrated and Cointegrated Attention use channel-independent (CI) modeling, ② Integrated Attention uses channel-dependent (CD) modeling while Cointegrated Attention uses CI, ③ Integrated Attention uses CI while Cointegrated Attention uses CD, and ④ both use CD modeling. The results in Tab. 7 show that modeling short-term inter-variates relationships can lead to severe spurious regression. CI modeling generally outperforms CD in scenarios with fewer channels (e.g., Weather), while CD excels when the number of channels is large (e.g., Traffic). This aligns with recent findings that inter-channel dependencies become increasingly important as the number of channels grows. We attribute this to the model's ability to extract potential long-term stable relationships from non-stationary sequences when more channels are present, thereby improving both forecasting accuracy and robustness.

| Case | Integrated Attention | Cointegrated Attention | Weather | | Solar | | Electricity | | Traffic | |
|---|---|---|---|---|---|---|---|---|---|---|
| | CI or CD | CI or CD | MSE | MAE | MSE | MAE | MSE | MAE | MSE | MAE |
| ① | CI | CI | **0.218** | **0.259** | 0.183 | 0.243 | 0.157 | 0.252 | 0.387 | 0.276 |
| ② | CD | CI | 0.222 | 0.262 | 0.184 | 0.247 | 0.160 | 0.255 | 0.387 | 0.280 |
| ③ | CI | CD | **0.218** | **0.259** | **0.181** | **0.239** | **0.149** | **0.246** | **0.360** | **0.255** |
| ④ | CD | CD | 0.222 | 0.263 | 0.183 | 0.247 | 0.156 | 0.254 | 0.376 | 0.269 |

Table 7: Ablation on modeling approaches for Integrated Attention and Cointegrated Attention. "CI" denotes channel independent and "CD" denotes channel-dependent modeling.

# 6 NON-STATIONARITY AND DEPENDENCY MODELING ANALYSIS

**Intra-variate Modeling.** As shown in Fig. 4a, when non-stationarity is retained, the attention map in the temporal dimension diverges, with the model focusing on multiple patches across a broader time span. However, after removing non-stationarity, the attention map becomes more concentrated on adjacent time steps, aligning with the causal nature of time series, where closer time steps are usually more correlated. Non-stationarity may cause the model to mistake distant similarities for causality. By eliminating non-stationarity, the model better captures short-term variations and local dependencies, enhancing its robustness and interpretability in handling complex time series data.

**Inter-variate Modeling.** Fig. 4b shows that removing non-stationarity narrows the model's attention to a few inter-variate dependencies, while retaining non-stationarity enables the capture of more diverse and richer relationships. Non-stationary sequences help the model identify cointegration, revealing hidden equilibrium mechanisms in multivariate time series. Preserving non-stationarity enhances the model's ability to express complex inter-variate dependencies. Additionally, Fig. 4c shows the impact of different patch downsampling rates on performance. For datasets with more

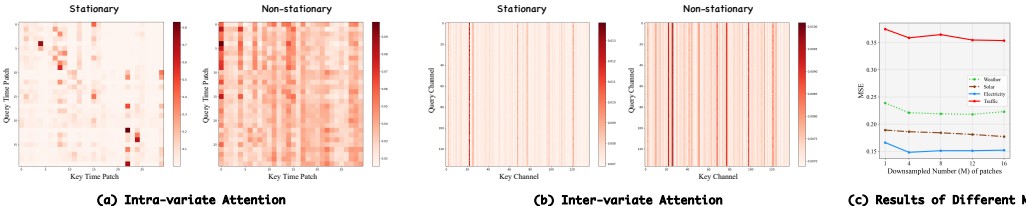

Figure 4: (a) Comparison of intra-variate attention maps under stationary and non-stationary conditions for different patches in the Electricity dataset. (b) Comparison of inter-variate attention maps between different variates under stationary and non-stationary conditions in the Solar dataset. (c) Impact of varying the number of downsampled patches $M$ on forecasting performance across different datasets. See Tab. 16 for full results.

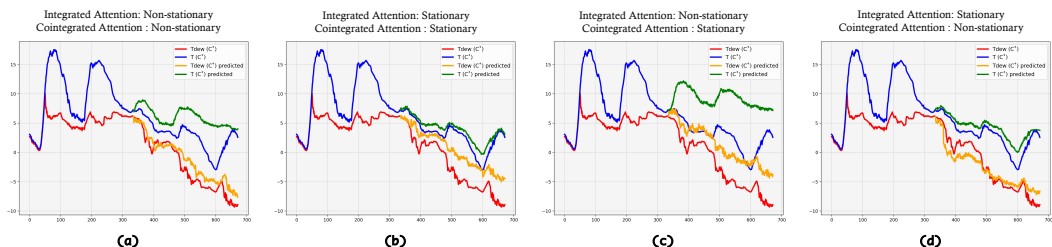

Figure 5: Visualization of the effect of retaining or removing non-stationarity in Integrated Attention and Cointegrated Attention on the Weather dataset for temperature ($T$) and dew point temperature ($T_{\text{dew}}$). (a) Both Integrated and Cointegrated Attention retain non-stationarity. (b) Both remove non-stationarity. (c) Only Integrated Attention retains non-stationarity. (d) Only Cointegrated Attention retains non-stationarity.

channels and stronger cointegration (e.g., Solar and Traffic), increasing downsampled patches initially improves predictions by preserving long-term features. However, too much downsampling adds computational cost and negatively affects smaller-channel datasets (e.g., Weather), so we carefully balanced downsampling rates based on dataset characteristics, as detailed in Tab. 9.

**Real Case of Weather Forecast.** Given the strong interrelationships between weather variables, we analyze temperature $T$ and dew point temperature $T_{\text{dew}}$ from the Weather dataset. Dew point measures atmospheric moisture and is typically closely linked to temperature. Without external influences, such as water vapor or heat sources, the difference between temperature and dew point is minimal, showing long-term cointegration. However, temperature tends to exhibit more short-term fluctuations due to external factors like sunlight and weather systems. As shown in Fig. 5, the results demonstrate that spurious regressions can only be avoided by eliminating non-stationarity during short-term modeling, while preserving it during long-term dependency modeling to capture the underlying cointegration between variables.

## 7 CONCLUSION

In this paper, we address the dual challenges of non-stationarity in multivariate time series forecasting, specifically focusing on its distinct impacts on short-term and long-term modeling. To this end, we propose TimeBridge, a novel framework that bridges the gap between non-stationarity and dependency modeling. By employing Integrated Attention to mitigate short-term non-stationarity and Cointegrated Attention to preserve long-term dependencies, TimeBridge effectively captures both local dynamics and long-term cointegration. Comprehensive experiments across diverse datasets demonstrate that TimeBridge consistently achieves state-of-the-art performance in both short-term and long-term forecasting tasks. Moreover, its exceptional performance on the CSI 500 and S&P 500 indices underscores its robustness and adaptability to complex real-world financial scenarios. Our work paves the way for further exploration of models that balance the nuanced effects of non-stationarity, offering a promising direction for advancing multivariate time series forecasting.

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

# A  TIME SERIES INTEGRATION AND COINTEGRATION ANALYSIS

## A.1  INTEGRATION AND ADF TEST

A time series is said to be integrated of order $k$, denoted as $I(k)$, if it becomes stationary after differencing $k$ times. For instance, a series $X_t$ is $I(1)$ if its first difference $\Delta X_t = X_t - X_{t-1}$ is stationary. To test for non-stationarity, the Augmented Dickey-Fuller (ADF) test (Mushtaq, 2011) is commonly used. It examines the null hypothesis that a unit root is present, indicating non-stationarity:

$$\Delta X_t = \alpha + \beta t + \gamma X_{t-1} + \sum_{i=1}^{p} \delta_i \Delta X_{t-i} + \epsilon_t$$

Here, $\Delta X_t$ is the differenced series, $\gamma$ is the coefficient on the lagged series, and $\epsilon_t$ is the error term. Rejecting the null hypothesis ($\gamma = 0$) indicates stationarity, while failing to reject it implies non-stationarity. Non-stationary data can lead to spurious regressions, where unrelated temporal intervals appear to be correlated due to common trends. We report the ADF test results in Tab. 8.

## A.2  COINTEGRATION AND EG TEST

Cointegration occurs when two or more non-stationary series move together over time, maintaining a stable, long-term relationship. For example, if $X_t$ and $Y_t$ are both $I(1)$, they are cointegrated if there exists a stationary linear combination, $Z_t = X_t - \beta Y_t$. This indicates a shared stochastic trend. The Engle-Granger (EG) test (Bilgili, 1998) for cointegration involves two steps:

1. Estimate Long-term Relationship. Regress $X_t$ on $Y_t$ using Ordinary Least Squares (OLS):

$$X_t = \alpha + \beta Y_t + \epsilon_t,$$

   where $\epsilon_t$ are the residuals.

2. ADF Test on Residuals. Apply the ADF test to $\epsilon_t$:

$$\Delta \epsilon_t = \gamma \epsilon_{t-1} + \sum_{i=1}^{p} \delta_i \Delta \epsilon_{t-i} + \nu_t$$

   If the residuals are stationary, $X_t$ and $Y_t$ are cointegrated.

Cointegration is vital for capturing long-term relationships between variables, providing a robust foundation for multivariate time series modeling. Ignoring cointegration can result in models that miss significant underlying connections, reducing forecasting accuracy and reliability. We report the EG test results in Tab. 8.

# B  METRICS

## B.1  LONG-TERM FORECASTING

We use Mean Squared Error (MSE) and Mean Absolute Error (MAE) as evaluation metrics. Given the ground truth values $\mathbf{X}_i$ and the predicted values $\hat{\mathbf{X}}_i$, these metrics are defined as follows:

$$\text{MSE} = \frac{1}{N} \sum_{i=1}^{N} (\mathbf{X}_i - \hat{\mathbf{X}}_i)^2, \quad \text{MAE} = \frac{1}{N} \sum_{i=1}^{N} |\mathbf{X}_i - \hat{\mathbf{X}}_i|,$$

where $N$ is the total number of predictions.

## B.2 SHORT-TERM FORECASTING

We use MAE (the same as defined above), Mean Absolute Percentage Error (MAPE), and Root Mean Squared Error (RMSE) to evaluate the performance. These metrics are defined as follows:

$$\text{MAPE} = \frac{1}{N} \sum_{i=1}^{N} \left| \frac{\mathbf{X}_i - \hat{\mathbf{X}}_i}{\mathbf{X}_i} \right| \times 100, \quad \text{RMSE} = \sqrt{\frac{1}{N} \sum_{i=1}^{N} (\mathbf{X}_i - \hat{\mathbf{X}}_i)^2}.$$

## B.3 FINANCIAL FORECASTING

We use six widely recognized metrics to assess the overall performance of each method: Annual Return Ratio (ARR), Annual Volatility (AVol), Maximum Drawdown (MDD), Annual Sharpe Ratio (ASR), Calmar Ratio (CR), and Information Ratio (IR). Lower absolute values of AVol and MDD, coupled with higher values of ARR, ASR, CR, and IR, indicate better performance.

- **ARR** quantifies the percentage increase or decrease in the value of an investment over a year.
$$\text{ARR} = (1 + \text{Total Return})^{\frac{1}{n}} - 1.$$

- **AVol** measures the volatility of an investment's returns over the course of a year. $R_p$ denotes the daily return of the portfolio.
$$\text{AVol} = \sqrt{\text{Var}(R_p)}.$$

- **MDD** indicates the maximum decline from a peak to a trough in the value of an investment.
$$\text{MDD} = -\max\left( \frac{p_{peak} - p_{trough}}{p_{peak}} \right).$$

- **ASR** reflects the risk-adjusted return of an investment over a year.
$$\text{ASR} = \frac{\text{ARR}}{\text{AVol}}.$$

- **CR** compares the average annual return of an investment to its maximum drawdown.
$$\text{CR} = \frac{\text{ARR}}{|\text{MDD}|}.$$

- **IR** evaluates the excess return of an investment relative to a benchmark, adjusted for its volatility. $R_b$ is the daily return of the market index.
$$\text{IR} = \frac{\text{mean}(R_p - R_b)}{\text{std}(R_p - R_b)}.$$

## C DATASETS

We conduct extensive experiments on eight widely-used time series datasets for long-term forecasting. Additionally, we use the PeMS datasets for short-term forecasting and the CSI 500 and S&P 500 indices for financial forecasting. We report the statistics in Tab. 8. Detailed descriptions of these datasets are as follows:

(1) **ETT** (Electricity Transformer Temperature) dataset (Zhou et al., 2021) encompasses temperature and power load data from electricity transformers in two regions of China, spanning from 2016 to 2018. This dataset has two granularity levels: ETTh (hourly) and ETTm (15 minutes).

(2) **Weather** dataset (Wu et al., 2023) captures 21 distinct meteorological indicators in Germany, meticulously recorded at 10-minute intervals throughout 2020. Key indicators in this dataset include air temperature, visibility, among others, offering a comprehensive view of the weather dynamics.

(3) **Electricity** dataset (Wu et al., 2023) features hourly electricity consumption records in kilowatt-hours (kWh) for 321 clients. Sourced from the UCL Machine Learning Repository, this dataset covers the period from 2012 to 2014, providing valuable insights into consumer electricity usage patterns.

(4) **Traffic** dataset (Wu et al., 2023) includes data on hourly road occupancy rates, gathered by 862 detectors across the freeways of the San Francisco Bay area. This dataset, covering the years 2015 to 2016, offers a detailed snapshot of traffic flow and congestion.

(5) **Solar-Energy** dataset (Lai et al., 2018) contains solar power production data recorded every 10 minutes throughout 2006 from 137 photovoltaic (PV) plants in Alabama.

(6) **PeMS** dataset (Liu et al., 2022a) comprises four public traffic network datasets (PeMS03, PeMS04, PeMS07, and PeMS08), constructed from the Caltrans Performance Measurement System (PeMS) across four districts in California. The data is aggregated into 5-minute intervals, resulting in 12 data points per hour and 288 data points per day.

(7) **CSI 500**[1] contains 502 stocks listed on the Shanghai and Shenzhen stock exchanges in China from 2018 to 2023, including close, open, high, low, volume and turnover data.

(8) **S&P 500**[2] contains 487 stocks representing diverse sectors within the U.S. economy from 2018 to 2023, including close, open, high, low and volume data.

| Tasks | Dataset | Dim | Prediction Length | Dataset Size | Frequency | ADF[†] | EG[‡] |
|---|---|---|---|---|---|---|---|
| | ETTm1 | 7 | $\{96, 192, 336, 720\}$ | $(34465, 11521, 11521)$ | 15 min | $-14.98$ | 20 |
| | ETTm2 | 7 | $\{96, 192, 336, 720\}$ | $(34465, 11521, 11521)$ | 15 min | $-5.66$ | 17 |
| | ETTh1 | 7 | $\{96, 192, 336, 720\}$ | $(8545, 2881, 2881)$ | 15 min | $-5.91$ | 11 |
| Long-term | ETTh2 | 7 | $\{96, 192, 336, 720\}$ | $(8545, 2881, 2881)$ | 15 min | $-4.13$ | 10 |
| Forecasting | Electricity | 321 | $\{96, 192, 336, 720\}$ | $(18317, 2633, 5261)$ | 1 hour | $-8.44$ | 39567 |
| | Traffic | 862 | $\{96, 192, 336, 720\}$ | $(12185, 1757, 3509)$ | 1 hour | $-15.02$ | 354627 |
| | Weather | 21 | $\{96, 192, 336, 720\}$ | $(36792, 5271, 10540)$ | 10 min | $-26.68$ | 77 |
| | Solar-Energy | 137 | $\{96, 192, 336, 720\}$ | $(36601, 5161, 10417)$ | 10 min | $-37.23$ | 8373 |
| | PeMS03 | 358 | 12 | $(15617, 5135, 5135)$ | 5 min | $-19.05$ | - |
| Short-term | PeMS04 | 307 | 12 | $(10172, 3375, 3375)$ | 5 min | $-15.66$ | - |
| Forecasting | PeMS07 | 883 | 12 | $(16911, 5622, 5622)$ | 5 min | $-20.60$ | - |
| | PeMS08 | 170 | 12 | $(10690, 3548, 265)$ | 5 min | $-16.04$ | - |
| Financial | CSI 500 | 502 | 1 | $(943, 242, 242)$ | 1 day | $-3.06$ | - |
| Forecasting | S&P 500 | 487 | 1 | $(1008, 251, 249)$ | 1 day | $-2.80$ | - |

† Augmented Dickey-Fuller (ADF) Test: A smaller ADF test result indicates a more stationary time series data.
‡ Engle-Granger (EG) Test: A bigger EG test result indicates the data contains more cointegration relationships.

Table 8: Dataset detailed descriptions. "Dataset Size" denotes the total number of time points in (Train, Validation, Test) split respectively. "Prediction Length" denotes the future time points to be predicted. "Frequency" denotes the sampling interval of time points.

To further illustrate the degree of non-stationarity in the datasets, we conduct additional experiments using a Random Walk series (representing maximum non-stationarity) and Gaussian white noise (representing near-stationarity). The Random Walk series is generated using the formula $X_t = X_{t-1} + \epsilon_t$ with $\epsilon_t \sim \mathcal{N}(0, 1)$, where we set $t = 10,000$ and simulate 100 iterations. The average ADF value for the Random Walk series is -1.53, indicating a high degree of non-stationarity. In contrast, for the Gaussian white noise series, generated as $X_t \sim \mathcal{N}(0, 1)$ with the same settings, the average ADF value is -97.54, indicating strong stationarity. Comparing these results with those in Tab. 8, we can see that most datasets exhibit significant non-stationarity, especially the ETT, CSI 500, and S&P 500 datasets.

---

[1]https://cn.investing.com/indices/china-securities-500
[2]https://hk.finance.yahoo.com/quote/%5EGSPC/history/

To analyze cointegration, we conducted the Engle-Granger (EG) test on all eight datasets of long-term forecasting. The results indicate that datasets with more channels tend to exhibit more extensive cointegration relationships. This is particularly evident in datasets like Electricity and Traffic, which show significantly higher EG test values, reflecting a greater abundance of long-term equilibrium relationships among variates. For these high-dimensional datasets, effectively modeling the intricate cointegration structures is crucial, as neglecting these long-term dependencies can result in suboptimal predictions.

## D    IMPLEMENTATION DETAILS

All experiments are implemented in PyTorch (Paszke et al., 2019) and conducted on two NVIDIA RTX 3090 24GB GPUs. We use the Adam optimizer (Kingma, 2014). The batch size is set to 16 for the Electricity and Traffic datasets, and 32 for all other datasets. All models are trained for 10 epochs. Tab. 9 provides detailed hyperparameter settings for each dataset. For the four ETT datasets, the relatively small number of channels results in less pronounced long-term cointegration relationships, as evidenced by the low EG test results in Tab. 8. Therefore, we focus on modeling short-term intra-variate variations only.

|  | Num. of Integrated | Num. of Cointegrated | $N$ | $M$ | lr | d_model | d_ff |
|---|---|---|---|---|---|---|---|
| ETTh1 | 2 | 0 | 30 | – | 1e-4 | 128 | 256 |
| ETTh2 | 2 | 0 | 30 | – | 1e-4 | 64 | 128 |
| ETTm1 | 2 | 0 | 30 | – | 1e-4 | 32 | 128 |
| ETTm2 | 2 | 0 | 30 | – | 1e-4 | 32 | 64 |
| Weather | 1 | 1 | 30 | 12 | 1e-4 | 128 | 128 |
| Solar | 1 | 1 | 30 | 12 | 5e-4 | 128 | 128 |
| Electricity | 2 | 1 | 30 | 4 | 5e-4 | 512 | 512 |
| Traffic | 2 | 2 | 30 | 4 | 1e-3 | 512 | 512 |

Table 9: Hyperparameter settings for different datasets. "$N$" denotes the number of patches. "$M$" denotes the number of patches after the patch downsampling block. "lr" denotes the learning rate. "d_model" and "d_ff" denote the model dimension of attention layers and feed-forward layers, respectively.

## E    FULL RESULTS

### E.1    MAIN EXPERIMENTS

Tab. 10 and Tab. 11 present the full results for long-term forecasting, including both the original results from their respective papers and those obtained through hyperparameter search. The hyperparameter search process involved exploring input lengths $I \in \{96, 192, 336, 512, 720\}$, learning rates from $10^{-5}$ to 0.05, encoder layers from 1 to 5, $d_{model}$ values from 16 to 512, and training epochs from 10 to 100. In both settings, TimeBridge consistently achieved the best performance, demonstrating its effectiveness and robustness. Additionally, for financial forecasting, we included three additional strong baselines: ALSTM (Qin et al., 2017), GRU (Chung et al., 2014), and TRA (Lin et al., 2021). The results in Tab. 12 show that TimeBridge continues to outperform these methods, further validating its superiority.

### E.2    ABLATION STUDIES

We present the full results of the ablation studies discussed in the main text. Tab. 13 provides the complete results of the ablation on removing non-stationarity in both Integrated and Cointegrated Attention. Tab. 14 reports the full results on the impact and order of Integrated and Cointegrated Attention, with an illustrative visualization in Fig. 6. Additionally, Tab. 15 shows the results of abla-

| Models | TimeBridge (Ours) | | iTransformer (2024a) | | PDF (2024) | | TimeMixer (2024) | | PatchTST (2023) | | Crossformer (2023) | | FEDformer (2022) | | ModernTCN (2024) | | MICN (2022) | | TimesNet (2023) | | DLinear (2023) | |
|---|---|---|---|---|---|---|---|---|---|---|---|---|---|---|---|---|---|---|---|---|---|---|---|
| Metric | MSE | MAE | MSE | MAE | MSE | MAE | MSE | MAE | MSE | MAE | MSE | MAE | MSE | MAE | MSE | MAE | MSE | MAE | MSE | MAE | MSE | MAE |
| ETTm1 96 | 0.297 | 0.353 | 0.334 | 0.368 | 0.280 | 0.335 | 0.320 | 0.357 | 0.293 | 0.346 | 0.316 | 0.373 | 0.379 | 0.419 | 0.292 | 0.346 | 0.316 | 0.362 | 0.338 | 0.375 | 0.299 | 0.343 |
| 192 | 0.333 | 0.375 | 0.377 | 0.391 | 0.317 | 0.359 | 0.361 | 0.381 | 0.333 | 0.370 | 0.377 | 0.411 | 0.426 | 0.441 | 0.332 | 0.368 | 0.363 | 0.390 | 0.374 | 0.387 | 0.335 | 0.365 |
| 336 | 0.366 | 0.399 | 0.426 | 0.420 | 0.354 | 0.382 | 0.390 | 0.404 | 0.369 | 0.392 | 0.431 | 0.442 | 0.445 | 0.459 | 0.365 | 0.391 | 0.408 | 0.426 | 0.410 | 0.411 | 0.369 | 0.386 |
| 720 | 0.414 | 0.423 | 0.491 | 0.459 | 0.405 | 0.413 | 0.454 | 0.441 | 0.416 | 0.420 | 0.600 | 0.547 | 0.543 | 0.490 | 0.416 | 0.417 | 0.481 | 0.476 | 0.478 | 0.450 | 0.425 | 0.421 |
| Avg. | 0.353 | 0.388 | 0.407 | 0.410 | 0.339 | 0.372 | 0.381 | 0.395 | 0.353 | 0.382 | 0.431 | 0.443 | 0.448 | 0.452 | 0.351 | 0.381 | 0.392 | 0.414 | 0.400 | 0.406 | 0.357 | 0.379 |
| ETTm2 96 | 0.158 | 0.249 | 0.180 | 0.264 | 0.162 | 0.253 | 0.175 | 0.258 | 0.166 | 0.256 | 0.275 | 0.358 | 0.203 | 0.287 | 0.166 | 0.256 | 0.203 | 0.287 | 0.187 | 0.267 | 0.167 | 0.260 |
| 192 | 0.215 | 0.291 | 0.250 | 0.309 | 0.219 | 0.291 | 0.237 | 0.299 | 0.223 | 0.296 | 0.345 | 0.400 | 0.269 | 0.328 | 0.222 | 0.293 | 0.262 | 0.326 | 0.249 | 0.309 | 0.224 | 0.303 |
| 336 | 0.263 | 0.323 | 0.311 | 0.348 | 0.270 | 0.326 | 0.298 | 0.340 | 0.274 | 0.329 | 0.657 | 0.528 | 0.325 | 0.366 | 0.272 | 0.324 | 0.305 | 0.353 | 0.321 | 0.351 | 0.281 | 0.342 |
| 720 | 0.348 | 0.376 | 0.412 | 0.407 | 0.358 | 0.380 | 0.391 | 0.396 | 0.362 | 0.385 | 1.208 | 0.753 | 0.421 | 0.415 | 0.351 | 0.381 | 0.389 | 0.407 | 0.408 | 0.403 | 0.397 | 0.421 |
| Avg. | 0.246 | 0.310 | 0.288 | 0.332 | 0.252 | 0.313 | 0.275 | 0.323 | 0.256 | 0.317 | 0.621 | 0.510 | 0.305 | 0.349 | 0.253 | 0.314 | 0.290 | 0.343 | 0.291 | 0.333 | 0.267 | 0.332 |
| ETTh1 96 | 0.358 | 0.392 | 0.386 | 0.405 | 0.357 | 0.388 | 0.375 | 0.400 | 0.370 | 0.400 | 0.405 | 0.426 | 0.376 | 0.419 | 0.368 | 0.394 | 0.398 | 0.427 | 0.384 | 0.402 | 0.375 | 0.399 |
| 192 | 0.388 | 0.411 | 0.441 | 0.436 | 0.397 | 0.412 | 0.429 | 0.421 | 0.413 | 0.429 | 0.419 | 0.444 | 0.420 | 0.448 | 0.405 | 0.413 | 0.430 | 0.453 | 0.436 | 0.429 | 0.405 | 0.416 |
| 336 | 0.401 | 0.419 | 0.487 | 0.458 | 0.409 | 0.422 | 0.484 | 0.458 | 0.422 | 0.440 | 0.440 | 0.461 | 0.459 | 0.465 | 0.391 | 0.412 | 0.440 | 0.460 | 0.491 | 0.469 | 0.439 | 0.443 |
| 720 | 0.447 | 0.458 | 0.503 | 0.491 | 0.432 | 0.455 | 0.498 | 0.482 | 0.447 | 0.468 | 0.519 | 0.524 | 0.506 | 0.507 | 0.450 | 0.461 | 0.491 | 0.509 | 0.521 | 0.500 | 0.472 | 0.490 |
| Avg. | 0.399 | 0.420 | 0.454 | 0.447 | 0.399 | 0.419 | 0.447 | 0.440 | 0.413 | 0.434 | 0.446 | 0.464 | 0.440 | 0.460 | 0.404 | 0.420 | 0.440 | 0.462 | 0.458 | 0.450 | 0.423 | 0.437 |
| ETTh2 96 | 0.295 | 0.354 | 0.297 | 0.349 | 0.272 | 0.333 | 0.289 | 0.341 | 0.274 | 0.337 | 0.628 | 0.563 | 0.346 | 0.388 | 0.263 | 0.332 | 0.332 | 0.377 | 0.340 | 0.374 | 0.289 | 0.353 |
| 192 | 0.351 | 0.389 | 0.380 | 0.400 | 0.335 | 0.375 | 0.372 | 0.392 | 0.314 | 0.382 | 0.703 | 0.624 | 0.429 | 0.439 | 0.320 | 0.374 | 0.422 | 0.441 | 0.402 | 0.414 | 0.383 | 0.418 |
| 336 | 0.351 | 0.397 | 0.428 | 0.432 | 0.325 | 0.377 | 0.386 | 0.414 | 0.329 | 0.384 | 0.827 | 0.675 | 0.496 | 0.487 | 0.313 | 0.376 | 0.447 | 0.474 | 0.452 | 0.452 | 0.448 | 0.465 |
| 720 | 0.388 | 0.436 | 0.427 | 0.445 | 0.375 | 0.417 | 0.412 | 0.434 | 0.379 | 0.422 | 1.181 | 0.840 | 0.463 | 0.474 | 0.392 | 0.433 | 0.442 | 0.467 | 0.462 | 0.468 | 0.605 | 0.551 |
| Avg. | 0.346 | 0.394 | 0.383 | 0.407 | 0.327 | 0.376 | 0.364 | 0.395 | 0.324 | 0.381 | 0.835 | 0.675 | 0.434 | 0.447 | 0.322 | 0.379 | 0.411 | 0.440 | 0.414 | 0.427 | 0.431 | 0.447 |
| Weather 96 | 0.143 | 0.192 | 0.174 | 0.214 | 0.147 | 0.194 | 0.163 | 0.209 | 0.149 | 0.198 | 0.153 | 0.217 | 0.217 | 0.296 | 0.149 | 0.200 | 0.161 | 0.229 | 0.172 | 0.220 | 0.176 | 0.237 |
| 192 | 0.185 | 0.235 | 0.221 | 0.254 | 0.192 | 0.239 | 0.208 | 0.250 | 0.194 | 0.241 | 0.197 | 0.269 | 0.276 | 0.336 | 0.196 | 0.245 | 0.220 | 0.281 | 0.219 | 0.261 | 0.220 | 0.282 |
| 336 | 0.237 | 0.277 | 0.278 | 0.296 | 0.244 | 0.282 | 0.251 | 0.287 | 0.245 | 0.282 | 0.495 | 0.515 | 0.339 | 0.380 | 0.238 | 0.277 | 0.278 | 0.331 | 0.280 | 0.306 | 0.265 | 0.319 |
| 720 | 0.307 | 0.330 | 0.358 | 0.349 | 0.318 | 0.330 | 0.339 | 0.341 | 0.314 | 0.334 | 0.526 | 0.542 | 0.403 | 0.428 | 0.314 | 0.334 | 0.311 | 0.356 | 0.365 | 0.359 | 0.323 | 0.362 |
| Avg. | 0.218 | 0.259 | 0.258 | 0.279 | 0.225 | 0.261 | 0.240 | 0.271 | 0.226 | 0.264 | 0.343 | 0.386 | 0.309 | 0.360 | 0.224 | 0.264 | 0.243 | 0.299 | 0.259 | 0.287 | 0.246 | 0.300 |
| Electricity 96 | 0.118 | 0.218 | 0.148 | 0.240 | 0.127 | 0.219 | 0.153 | 0.247 | 0.129 | 0.222 | 0.187 | 0.283 | 0.183 | 0.297 | 0.129 | 0.226 | 0.164 | 0.269 | 0.168 | 0.272 | 0.140 | 0.237 |
| 192 | 0.142 | 0.237 | 0.162 | 0.253 | 0.145 | 0.237 | 0.166 | 0.256 | 0.147 | 0.240 | 0.258 | 0.330 | 0.195 | 0.308 | 0.143 | 0.239 | 0.177 | 0.285 | 0.184 | 0.289 | 0.153 | 0.249 |
| 336 | 0.156 | 0.252 | 0.178 | 0.269 | 0.162 | 0.255 | 0.185 | 0.277 | 0.163 | 0.259 | 0.323 | 0.369 | 0.212 | 0.313 | 0.161 | 0.259 | 0.193 | 0.304 | 0.198 | 0.300 | 0.169 | 0.267 |
| 720 | 0.179 | 0.278 | 0.225 | 0.317 | 0.200 | 0.290 | 0.225 | 0.310 | 0.197 | 0.290 | 0.404 | 0.423 | 0.231 | 0.343 | 0.191 | 0.286 | 0.212 | 0.321 | 0.220 | 0.320 | 0.203 | 0.301 |
| Avg. | 0.149 | 0.246 | 0.178 | 0.270 | 0.159 | 0.250 | 0.182 | 0.272 | 0.159 | 0.253 | 0.293 | 0.351 | 0.205 | 0.315 | 0.156 | 0.253 | 0.187 | 0.295 | 0.192 | 0.295 | 0.166 | 0.264 |
| Traffic 96 | 0.340 | 0.240 | 0.395 | 0.268 | 0.351 | 0.238 | 0.462 | 0.285 | 0.360 | 0.249 | 0.512 | 0.290 | 0.562 | 0.349 | 0.368 | 0.253 | 0.519 | 0.309 | 0.593 | 0.321 | 0.410 | 0.282 |
| 192 | 0.343 | 0.250 | 0.417 | 0.276 | 0.374 | 0.248 | 0.473 | 0.296 | 0.379 | 0.256 | 0.523 | 0.297 | 0.562 | 0.346 | 0.379 | 0.261 | 0.537 | 0.315 | 0.617 | 0.336 | 0.423 | 0.287 |
| 336 | 0.363 | 0.257 | 0.433 | 0.283 | 0.386 | 0.253 | 0.498 | 0.296 | 0.392 | 0.264 | 0.530 | 0.300 | 0.570 | 0.323 | 0.397 | 0.270 | 0.534 | 0.313 | 0.629 | 0.336 | 0.436 | 0.296 |
| 720 | 0.393 | 0.271 | 0.467 | 0.302 | 0.421 | 0.278 | 0.506 | 0.313 | 0.432 | 0.286 | 0.573 | 0.313 | 0.596 | 0.368 | 0.440 | 0.291 | 0.577 | 0.325 | 0.640 | 0.350 | 0.466 | 0.315 |
| Avg. | 0.360 | 0.255 | 0.428 | 0.282 | 0.383 | 0.254 | 0.484 | 0.297 | 0.391 | 0.264 | 0.535 | 0.300 | 0.573 | 0.347 | 0.396 | 0.270 | 0.542 | 0.316 | 0.620 | 0.336 | 0.434 | 0.295 |
| Solar 96 | 0.161 | 0.224 | 0.203 | 0.237 | 0.179 | 0.246 | 0.189 | 0.259 | 0.190 | 0.273 | 0.181 | 0.240 | 0.209 | 0.330 | 0.202 | 0.263 | 0.190 | 0.250 | 0.285 | 0.330 | 0.289 | 0.377 |
| 192 | 0.177 | 0.237 | 0.233 | 0.261 | 0.205 | 0.265 | 0.222 | 0.283 | 0.204 | 0.302 | 0.196 | 0.252 | 0.274 | 0.400 | 0.223 | 0.279 | 0.225 | 0.270 | 0.309 | 0.342 | 0.319 | 0.397 |
| 336 | 0.188 | 0.244 | 0.248 | 0.273 | 0.210 | 0.270 | 0.231 | 0.292 | 0.212 | 0.293 | 0.216 | 0.243 | 0.338 | 0.439 | 0.241 | 0.292 | 0.250 | 0.301 | 0.335 | 0.365 | 0.352 | 0.415 |
| 720 | 0.197 | 0.252 | 0.249 | 0.275 | 0.225 | 0.281 | 0.223 | 0.285 | 0.221 | 0.310 | 0.220 | 0.256 | 0.365 | 0.459 | 0.247 | 0.292 | 0.323 | 0.362 | 0.346 | 0.355 | 0.356 | 0.412 |
| Avg. | 0.181 | 0.239 | 0.233 | 0.262 | 0.205 | 0.265 | 0.216 | 0.280 | 0.207 | 0.294 | 0.204 | 0.248 | 0.296 | 0.407 | 0.228 | 0.282 | 0.247 | 0.296 | 0.319 | 0.348 | 0.329 | 0.400 |

Table 10: Full results of long-term forecasting from the original papers. All results are averaged across four different prediction lengths: $O \in \{96, 192, 336, 720\}$. The best and second-best results are highlighted in **bold** and underlined, respectively.

tion on different modeling approaches for these attention mechanisms. Finally, Tab. 16 presents the results of varying the number of downsampled patches $M$ and its effect on forecasting performance.

# F STATISTICAL ANALYSIS

We repeat all experiments three times and report the standard deviations for both our model and the second-best baseline, along with the results of statistical significance tests. Tab. 17, Tab. 18, and Tab. 19 present the results for long-term forecasting, short-term forecasting, and financial forecasting, respectively.

# G VISUALIZATION

Fig. 7 visualizes short-term fluctuations and long-term cointegration across stock sectors. Fig. 8 provides additional examples of intra-variate attention maps comparing stationary and non-stationary conditions for different patches in the Electricity dataset. Fig. 9 shows further examples of inter-variate attention maps in the Solar dataset under both stationary and non-stationary conditions. Fig. 10, Fig. 11, Fig. 12, and Fig. 13 present long-term forecasting visualizations for Weather, Solar, Electricity, and Traffic datasets, respectively. We display the last 96 input steps based on each

model's optimal input length, along with the corresponding 96 predicted steps. Finally, Fig. 14 illustrates short-term forecasting for the PeMS03 dataset, where each model predicts 12 steps from a 96-step input.

| Models | | TimeBridge (Ours) | | iTransformer (2024a) | | PDF (2024) | | TimeMixer (2024) | | PatchTST (2023) | | Crossformer (2023) | | FEDformer (2022) | | ModernTCN (2024) | | MICN (2022) | | TimesNet (2023) | | DLinear (2023) | |
|---|---|---|---|---|---|---|---|---|---|---|---|---|---|---|---|---|---|---|---|---|---|---|---|
| Metric | | MSE | MAE | MSE | MAE | MSE | MAE | MSE | MAE | MSE | MAE | MSE | MAE | MSE | MAE | MSE | MAE | MSE | MAE | MSE | MAE | MSE | MAE |
| ETTm1 | 96 | 0.297 | 0.353 | 0.300 | 0.353 | **0.277** | **0.337** | 0.291 | 0.340 | 0.293 | 0.346 | 0.310 | 0.361 | 0.326 | 0.390 | 0.292 | 0.346 | 0.314 | 0.360 | 0.338 | 0.375 | 0.299 | 0.343 |
| | 192 | 0.333 | 0.375 | 0.345 | 0.382 | **0.316** | **0.364** | 0.327 | 0.365 | 0.333 | 0.370 | 0.363 | 0.402 | 0.365 | 0.415 | 0.332 | 0.368 | 0.359 | 0.387 | 0.371 | 0.387 | 0.335 | 0.365 |
| | 336 | 0.366 | 0.399 | 0.374 | 0.398 | **0.346** | **0.381** | 0.360 | 0.381 | 0.369 | 0.392 | 0.408 | 0.430 | 0.392 | 0.425 | 0.365 | 0.391 | 0.398 | 0.413 | 0.410 | 0.411 | 0.369 | 0.386 |
| | 720 | 0.414 | 0.423 | 0.429 | 0.430 | **0.402** | **0.409** | 0.415 | 0.417 | 0.416 | 0.420 | 0.600 | 0.547 | 0.446 | 0.458 | 0.416 | 0.417 | 0.459 | 0.464 | 0.478 | 0.450 | 0.425 | 0.421 |
| | Avg. | 0.353 | 0.388 | 0.362 | 0.391 | **0.335** | **0.373** | 0.348 | 0.375 | 0.353 | 0.382 | 0.420 | 0.435 | 0.382 | 0.422 | 0.351 | 0.381 | 0.383 | 0.406 | 0.400 | 0.406 | 0.357 | 0.379 |
| ETTm2 | 96 | **0.158** | **0.249** | 0.175 | 0.266 | 0.159 | 0.251 | 0.164 | 0.254 | 0.166 | 0.256 | 0.263 | 0.359 | 0.180 | 0.271 | 0.166 | 0.256 | 0.178 | 0.273 | 0.187 | 0.267 | 0.167 | 0.260 |
| | 192 | **0.215** | **0.291** | 0.242 | 0.312 | 0.217 | 0.292 | 0.223 | 0.295 | 0.223 | 0.296 | 0.345 | 0.400 | 0.252 | 0.318 | 0.222 | 0.293 | 0.245 | 0.316 | 0.249 | 0.309 | 0.224 | 0.303 |
| | 336 | **0.263** | **0.323** | 0.282 | 0.340 | 0.266 | 0.325 | 0.279 | 0.330 | 0.274 | 0.329 | 0.469 | 0.496 | 0.324 | 0.364 | 0.272 | 0.324 | 0.295 | 0.350 | 0.321 | 0.351 | 0.281 | 0.342 |
| | 720 | 0.348 | 0.376 | 0.378 | 0.398 | **0.345** | **0.375** | 0.359 | 0.383 | 0.362 | 0.385 | 0.996 | 0.750 | 0.410 | 0.420 | 0.351 | 0.381 | 0.389 | 0.406 | 0.497 | 0.403 | 0.397 | 0.421 |
| | Avg. | **0.246** | **0.310** | 0.269 | 0.329 | 0.247 | 0.311 | 0.256 | 0.315 | 0.256 | 0.317 | 0.518 | 0.501 | 0.292 | 0.343 | 0.253 | 0.314 | 0.277 | 0.336 | 0.291 | 0.333 | 0.267 | 0.332 |
| ETTh1 | 96 | 0.358 | 0.392 | 0.386 | 0.405 | **0.356** | **0.391** | 0.361 | **0.390** | 0.370 | 0.400 | 0.386 | 0.426 | 0.376 | 0.415 | 0.368 | 0.394 | 0.396 | 0.427 | 0.384 | 0.402 | 0.375 | 0.399 |
| | 192 | **0.388** | **0.411** | 0.424 | 0.440 | 0.390 | 0.413 | 0.409 | 0.414 | 0.413 | 0.429 | 0.413 | 0.442 | 0.423 | 0.446 | 0.405 | 0.413 | 0.430 | 0.453 | 0.557 | 0.436 | 0.405 | 0.416 |
| | 336 | 0.401 | 0.419 | 0.449 | 0.460 | 0.402 | 0.421 | 0.430 | 0.429 | 0.422 | 0.440 | 0.440 | 0.461 | 0.444 | 0.462 | **0.391** | **0.412** | 0.433 | 0.458 | 0.491 | 0.469 | 0.439 | 0.443 |
| | 720 | 0.447 | 0.458 | 0.495 | 0.487 | **0.432** | **0.455** | 0.445 | 0.460 | 0.447 | 0.468 | 0.519 | 0.524 | 0.469 | 0.492 | 0.450 | 0.461 | 0.474 | 0.508 | 0.521 | 0.500 | 0.472 | 0.490 |
| | Avg. | 0.399 | **0.420** | 0.439 | 0.448 | **0.395** | **0.420** | 0.411 | 0.423 | 0.413 | 0.434 | 0.440 | 0.463 | 0.428 | 0.454 | 0.404 | **0.420** | 0.433 | 0.462 | 0.458 | 0.450 | 0.423 | 0.437 |
| ETTh2 | 96 | 0.295 | 0.354 | 0.297 | 0.348 | 0.270 | 0.332 | 0.271 | **0.330** | 0.274 | 0.337 | 0.611 | 0.557 | 0.332 | 0.374 | **0.263** | 0.332 | 0.289 | 0.357 | 0.340 | 0.374 | 0.289 | 0.353 |
| | 192 | 0.351 | 0.389 | 0.371 | 0.403 | 0.334 | 0.375 | 0.317 | 0.402 | 0.314 | 0.382 | 0.703 | 0.624 | 0.407 | 0.446 | 0.320 | **0.374** | 0.409 | 0.438 | 0.402 | 0.414 | 0.383 | 0.418 |
| | 336 | 0.351 | 0.397 | 0.404 | 0.428 | 0.324 | 0.379 | 0.332 | 0.396 | 0.329 | 0.384 | 0.827 | 0.675 | 0.400 | 0.447 | **0.313** | **0.376** | 0.417 | 0.452 | 0.452 | 0.452 | 0.448 | 0.465 |
| | 720 | 0.388 | 0.436 | 0.424 | 0.444 | 0.375 | 0.417 | **0.342** | **0.408** | 0.379 | 0.422 | 1.094 | 0.775 | 0.412 | 0.469 | 0.392 | 0.433 | 0.426 | 0.473 | 0.462 | 0.468 | 0.605 | 0.551 |
| | Avg. | 0.346 | 0.394 | 0.374 | 0.406 | 0.326 | 0.376 | **0.316** | 0.384 | 0.324 | 0.381 | 0.809 | 0.658 | 0.388 | 0.434 | 0.322 | 0.379 | 0.385 | 0.430 | 0.414 | 0.427 | 0.431 | 0.447 |
| Weather | 96 | **0.143** | **0.192** | 0.159 | 0.208 | **0.143** | 0.193 | 0.147 | 0.197 | 0.149 | 0.198 | 0.146 | 0.212 | 0.217 | 0.296 | 0.149 | 0.200 | 0.161 | 0.226 | 0.172 | 0.220 | 0.152 | 0.237 |
| | 192 | **0.185** | **0.235** | 0.200 | 0.248 | 0.188 | 0.236 | 0.189 | 0.239 | 0.194 | 0.241 | 0.195 | 0.261 | 0.275 | 0.329 | 0.196 | 0.245 | 0.220 | 0.283 | 0.219 | 0.261 | 0.220 | 0.282 |
| | 336 | 0.237 | **0.277** | 0.253 | 0.289 | 0.240 | 0.279 | 0.241 | 0.280 | 0.245 | 0.282 | 0.252 | 0.311 | 0.339 | 0.377 | **0.238** | **0.277** | 0.275 | 0.328 | 0.280 | 0.306 | 0.265 | 0.319 |
| | 720 | **0.307** | 0.330 | 0.321 | 0.338 | 0.308 | **0.328** | 0.310 | 0.330 | 0.314 | 0.334 | 0.318 | 0.363 | 0.389 | 0.409 | 0.314 | 0.334 | 0.311 | 0.356 | 0.365 | 0.359 | 0.323 | 0.362 |
| | Avg. | **0.218** | **0.259** | 0.233 | 0.271 | 0.220 | 0.259 | 0.222 | 0.262 | 0.226 | 0.264 | 0.228 | 0.287 | 0.305 | 0.287 | 0.224 | 0.264 | 0.242 | 0.298 | 0.259 | 0.287 | 0.240 | 0.300 |
| Electricity | 96 | **0.118** | **0.218** | 0.138 | 0.237 | 0.126 | 0.220 | 0.129 | 0.224 | 0.129 | 0.222 | 0.135 | 0.237 | 0.183 | 0.297 | 0.129 | 0.226 | 0.159 | 0.267 | 0.168 | 0.272 | 0.140 | 0.237 |
| | 192 | 0.142 | 0.237 | 0.157 | 0.256 | 0.145 | 0.237 | **0.140** | **0.220** | 0.147 | 0.240 | 0.160 | 0.262 | 0.195 | 0.308 | 0.143 | 0.239 | 0.168 | 0.279 | 0.184 | 0.289 | 0.152 | 0.249 |
| | 336 | **0.156** | **0.252** | 0.167 | 0.264 | 0.159 | 0.252 | 0.161 | 0.255 | 0.163 | 0.259 | 0.182 | 0.282 | 0.212 | 0.313 | 0.161 | 0.259 | 0.196 | 0.308 | 0.198 | 0.300 | 0.169 | 0.267 |
| | 720 | **0.179** | **0.278** | 0.194 | 0.286 | 0.194 | 0.287 | 0.194 | 0.287 | 0.197 | 0.290 | 0.246 | 0.337 | 0.231 | 0.343 | 0.191 | 0.286 | 0.203 | 0.312 | 0.220 | 0.320 | 0.203 | 0.301 |
| | Avg. | **0.149** | **0.246** | 0.164 | 0.261 | 0.156 | 0.250 | 0.156 | **0.246** | 0.159 | 0.253 | 0.181 | 0.279 | 0.205 | 0.315 | 0.156 | 0.253 | 0.182 | 0.292 | 0.192 | 0.295 | 0.166 | 0.264 |
| Traffic | 96 | **0.340** | 0.240 | 0.363 | 0.265 | 0.350 | **0.239** | 0.360 | 0.249 | 0.360 | 0.249 | 0.512 | 0.282 | 0.562 | 0.349 | 0.368 | 0.253 | 0.508 | 0.301 | 0.593 | 0.321 | 0.410 | 0.282 |
| | 192 | **0.343** | 0.250 | 0.385 | 0.273 | 0.363 | **0.247** | 0.375 | 0.250 | 0.379 | 0.256 | 0.501 | 0.273 | 0.562 | 0.346 | 0.379 | 0.261 | 0.536 | 0.315 | 0.617 | 0.336 | 0.423 | 0.287 |
| | 336 | **0.363** | **0.257** | 0.396 | 0.277 | 0.376 | 0.258 | 0.385 | 0.270 | 0.392 | 0.264 | 0.507 | 0.279 | 0.570 | 0.323 | 0.397 | 0.270 | 0.525 | 0.310 | 0.629 | 0.336 | 0.436 | 0.296 |
| | 720 | **0.393** | **0.271** | 0.445 | 0.312 | 0.419 | 0.279 | 0.430 | 0.281 | 0.432 | 0.286 | 0.571 | 0.301 | 0.596 | 0.368 | 0.440 | 0.282 | 0.571 | 0.323 | 0.640 | 0.350 | 0.466 | 0.315 |
| | Avg. | **0.360** | **0.255** | 0.397 | 0.282 | 0.377 | 0.256 | 0.387 | 0.262 | 0.391 | 0.264 | 0.523 | 0.284 | 0.573 | 0.347 | 0.396 | 0.270 | 0.535 | 0.312 | 0.620 | 0.336 | 0.434 | 0.295 |
| Solar | 96 | **0.161** | 0.224 | 0.188 | 0.242 | 0.179 | 0.246 | 0.167 | **0.220** | 0.178 | 0.229 | 0.166 | 0.230 | 0.201 | 0.304 | 0.202 | 0.263 | 0.188 | 0.252 | 0.219 | 0.314 | 0.216 | 0.287 |
| | 192 | **0.177** | **0.237** | 0.193 | 0.258 | 0.205 | 0.265 | 0.187 | 0.249 | 0.189 | 0.246 | 0.186 | **0.237** | 0.237 | 0.337 | 0.223 | 0.279 | 0.215 | 0.280 | 0.231 | 0.322 | 0.244 | 0.305 |
| | 336 | **0.188** | 0.244 | 0.195 | 0.259 | 0.210 | 0.270 | 0.200 | 0.258 | 0.198 | 0.249 | 0.203 | **0.243** | 0.254 | 0.362 | 0.241 | 0.292 | 0.222 | 0.267 | 0.246 | 0.337 | 0.263 | 0.319 |
| | 720 | **0.197** | **0.252** | 0.223 | 0.281 | 0.225 | 0.281 | 0.215 | 0.250 | 0.209 | 0.256 | 0.210 | 0.256 | 0.280 | 0.397 | 0.247 | 0.292 | 0.226 | 0.264 | 0.280 | 0.363 | 0.264 | 0.324 |
| | Avg. | **0.181** | **0.239** | 0.200 | 0.260 | 0.205 | 0.265 | 0.192 | 0.244 | 0.194 | 0.245 | 0.191 | 0.242 | 0.243 | 0.350 | 0.228 | 0.282 | 0.213 | 0.266 | 0.244 | 0.334 | 0.247 | 0.309 |

Table 11: Full results of long-term forecasting of hyperparameter searching. All results are averaged across four different prediction lengths: $O \in \{96, 192, 336, 720\}$. The best and second-best results are highlighted in **bold** and underlined, respectively.

| Models | CSI 500 | | | | | | S&P 500 | | | | | |
|---|---|---|---|---|---|---|---|---|---|---|---|---|
| | ARR↑ | AVol↓ | MDD↓ | ASR↑ | CR↑ | IR↑ | ARR↑ | AVol↓ | MDD↓ | ASR↑ | CR↑ | IR↑ |
| BLSW (1988) | 0.110 | 0.227 | -0.155 | 0.485 | 0.710 | 0.446 | 0.199 | 0.318 | -0.223 | 0.626 | 0.892 | 0.774 |
| CSM (1993) | 0.015 | 0.229 | -0.179 | 0.066 | 0.084 | 0.001 | 0.099 | 0.250 | -0.139 | 0.396 | 0.712 | 0.584 |
| LSTM (1997) | -0.008 | 0.159 | -0.172 | -0.047 | -0.044 | -0.128 | 0.142 | 0.162 | -0.178 | 0.877 | 0.798 | 0.929 |
| ALSTM (2017) | 0.016 | 0.162 | -0.192 | 0.101 | 0.086 | 0.014 | 0.191 | 0.161 | -0.150 | 1.186 | 1.273 | 1.115 |
| GRU (2014) | -0.004 | 0.159 | -0.193 | -0.028 | -0.023 | -0.118 | 0.124 | 0.169 | -0.139 | 0.734 | 0.829 | 1.023 |
| Transformer (2017b) | 0.154 | 0.156 | -0.135 | 0.986 | 1.143 | 0.867 | 0.135 | 0.159 | -0.140 | 0.852 | 0.968 | 0.908 |
| TRA (2021) | 0.125 | 0.162 | -0.145 | 0.776 | 0.866 | 0.657 | 0.184 | 0.166 | -0.158 | 1.114 | 1.172 | 1.106 |
| PatchTST (2023) | 0.118 | **0.152** | -0.127 | 0.776 | 0.923 | 0.735 | 0.146 | 0.167 | -0.140 | 0.877 | 1.042 | 0.949 |
| iTransformer (2024a) | 0.214 | 0.168 | -0.164 | 1.276 | 1.309 | 1.173 | 0.159 | 0.170 | -0.139 | 0.941 | 1.150 | 0.955 |
| TimeMixer (2024) | 0.078 | 0.153 | **-0.114** | 0.511 | 0.685 | 0.385 | 0.254 | 0.162 | -0.131 | 1.568 | 1.938 | 1.448 |
| Crossformer (2023) | -0.039 | 0.163 | -0.217 | -0.238 | -0.179 | -0.350 | 0.284 | **0.159** | **-0.114** | 1.786 | **2.491** | 1.646 |
| TimeBridge | **0.285** | 0.203 | -0.196 | **1.405** | **1.453** | **1.317** | **0.326** | 0.169 | -0.142 | **1.927** | 2.298 | **1.842** |

Table 12: Full results for financial time series forecasting in CSI 500 and S&P 500 datasets.

| Integrated Attention + Norm? | Cointegrated Attention + Norm? | Length | Weather MSE | Weather MAE | Solar MSE | Solar MAE | Electricity MSE | Electricity MAE | Traffic MSE | Traffic MAE |
|---|---|---|---|---|---|---|---|---|---|---|
| × | × | 96 | 0.144 | 0.193 | 0.163 | 0.227 | 0.124 | 0.221 | 0.342 | 0.241 |
| | | 192 | 0.186 | 0.235 | 0.180 | 0.240 | 0.144 | 0.240 | 0.351 | 0.255 |
| | | 336 | 0.239 | 0.279 | 0.191 | 0.248 | 0.158 | 0.254 | 0.374 | 0.261 |
| | | 720 | 0.311 | 0.333 | 0.197 | 0.253 | 0.184 | 0.282 | 0.418 | 0.284 |
| | | *Avg.* | 0.220 | 0.260 | 0.183 | 0.242 | 0.153 | 0.249 | 0.371 | 0.260 |
| × | ✓ | 96 | 0.144 | 0.193 | 0.164 | 0.227 | 0.124 | 0.220 | 0.348 | 0.247 |
| | | 192 | 0.188 | 0.237 | 0.180 | 0.240 | 0.146 | 0.240 | 0.370 | 0.257 |
| | | 336 | 0.239 | 0.279 | 0.191 | 0.248 | 0.161 | 0.258 | 0.382 | 0.264 |
| | | 720 | 0.308 | 0.331 | 0.197 | 0.293 | 0.189 | 0.285 | 0.422 | 0.283 |
| | | *Avg.* | 0.220 | 0.260 | 0.183 | 0.252 | 0.155 | 0.251 | 0.381 | 0.263 |
| ✓ | × | 96 | 0.143 | 0.192 | 0.161 | 0.224 | 0.118 | 0.218 | 0.340 | 0.240 |
| | | 192 | 0.185 | 0.235 | 0.177 | 0.237 | 0.142 | 0.237 | 0.343 | 0.250 |
| | | 336 | 0.237 | 0.277 | 0.188 | 0.244 | 0.156 | 0.252 | 0.363 | 0.257 |
| | | 720 | 0.307 | 0.330 | 0.197 | 0.252 | 0.179 | 0.278 | 0.393 | 0.271 |
| | | *Avg.* | **0.218** | **0.259** | **0.181** | **0.239** | **0.149** | **0.246** | **0.360** | **0.255** |
| ✓ | ✓ | 96 | 0.143 | 0.193 | 0.163 | 0.227 | 0.123 | 0.219 | 0.343 | 0.241 |
| | | 192 | 0.186 | 0.236 | 0.180 | 0.239 | 0.144 | 0.239 | 0.367 | 0.254 |
| | | 336 | 0.238 | 0.278 | 0.191 | 0.247 | 0.159 | 0.256 | 0.379 | 0.262 |
| | | 720 | 0.307 | 0.330 | 0.197 | 0.253 | 0.185 | 0.284 | 0.405 | 0.277 |
| | | *Avg.* | 0.219 | **0.259** | 0.183 | 0.242 | 0.153 | 0.250 | 0.374 | 0.289 |

Table 13: Full results of ablation on the effect of removing non-stationarity in Integrated Attention and Cointegrated Attention. ✓ indicates the use of patch normalization to eliminate non-stationarity, while × means non-stationarity is retained.

| Integrated Attention Order | Cointegrated Attention Order | Length | Weather MSE | Weather MAE | Solar MSE | Solar MAE | Electricity MSE | Electricity MAE | Traffic MSE | Traffic MAE |
|---|---|---|---|---|---|---|---|---|---|---|
| 1 | × | 96 | 0.144 | 0.196 | 0.163 | 0.227 | 0.127 | 0.221 | 0.356 | 0.245 |
| | | 192 | 0.186 | 0.238 | 0.182 | 0.242 | 0.145 | 0.239 | 0.377 | 0.259 |
| | | 336 | 0.241 | 0.283 | 0.192 | 0.246 | 0.162 | 0.257 | 0.390 | 0.265 |
| | | 720 | 0.310 | 0.332 | 0.199 | 0.260 | 0.197 | 0.289 | 0.427 | 0.286 |
| | | *Avg.* | 0.220 | 0.262 | 0.184 | 0.244 | 0.158 | 0.252 | 0.388 | 0.264 |
| × | 1 | 96 | 0.147 | 0.200 | 0.161 | 0.240 | 0.127 | 0.227 | 0.348 | 0.252 |
| | | 192 | 0.191 | 0.242 | 0.195 | 0.259 | 0.155 | 0.254 | 0.358 | 0.262 |
| | | 336 | 0.242 | 0.283 | 0.198 | 0.268 | 0.173 | 0.273 | 0.367 | 0.267 |
| | | 720 | 0.308 | 0.331 | 0.209 | 0.273 | 0.203 | 0.299 | 0.401 | 0.278 |
| | | *Avg.* | 0.222 | 0.264 | 0.191 | 0.260 | 0.165 | 0.263 | 0.369 | 0.265 |
| 1 | 2 | 96 | 0.143 | 0.193 | 0.161 | 0.224 | 0.118 | 0.218 | 0.340 | 0.240 |
| | | 192 | 0.185 | 0.235 | 0.177 | 0.237 | 0.142 | 0.237 | 0.343 | 0.250 |
| | | 336 | 0.237 | 0.277 | 0.188 | 0.244 | 0.156 | 0.252 | 0.363 | 0.257 |
| | | 720 | 0.307 | 0.330 | 0.197 | 0.252 | 0.179 | 0.278 | 0.393 | 0.271 |
| | | *Avg.* | **0.218** | **0.259** | **0.181** | **0.239** | **0.149** | **0.246** | **0.360** | **0.255** |
| 2 | 1 | 96 | 0.148 | 0.199 | 0.174 | 0.237 | 0.130 | 0.225 | 0.370 | 0.274 |
| | | 192 | 0.193 | 0.243 | 0.187 | 0.251 | 0.147 | 0.240 | 0.386 | 0.279 |
| | | 336 | 0.245 | 0.284 | 0.195 | 0.258 | 0.165 | 0.262 | 0.394 | 0.276 |
| | | 720 | 0.320 | 0.336 | 0.203 | 0.262 | 0.199 | 0.291 | 0.432 | 0.295 |
| | | *Avg.* | 0.227 | 0.266 | 0.190 | 0.252 | 0.160 | 0.255 | 0.396 | 0.281 |

Table 14: Full results of ablation on the impact and order of Integrated Attention and Cointegrated Attention. "Order" specifies the sequence, with lower numbers indicating earlier placement. × indicates the component is removed.

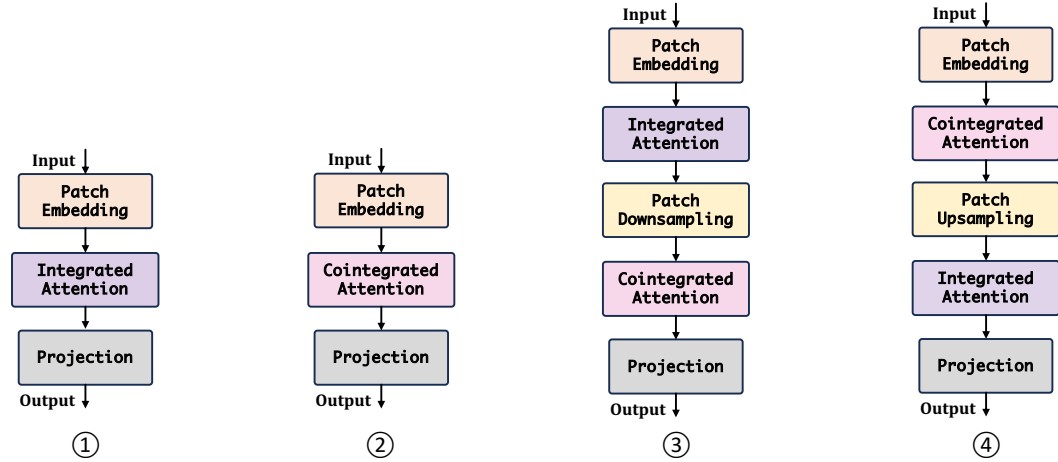

Figure 6: Illustration of the impact and order of Integrated Attention and Cointegrated Attention in Tab. 14: ① Integrated Attention only, ② Cointegrated Attention only, ③ Integrated Attention followed by Cointegrated Attention, and ④ Cointegrated Attention followed by Integrated Attention, with patch downsampling replaced by upsampling.

| Integrated Attention | Cointegrated Attention | | Weather | | Solar | | Electricity | | Traffic | |
|---|---|---|---|---|---|---|---|---|---|---|
| CI or CD | CI or CD | Length | MSE | MAE | MSE | MAE | MSE | MAE | MSE | MAE |
| CI | CI | 96 | 0.144 | 0.192 | 0.163 | 0.227 | 0.125 | 0.221 | 0.358 | 0.260 |
| | | 192 | 0.185 | 0.236 | 0.180 | 0.240 | 0.144 | 0.242 | 0.373 | 0.271 |
| | | 336 | 0.237 | 0.278 | 0.191 | 0.247 | 0.161 | 0.255 | 0.391 | 0.282 |
| | | 720 | 0.307 | 0.330 | 0.197 | 0.258 | 0.196 | 0.288 | 0.425 | 0.292 |
| | | *Avg.* | **0.218** | **0.259** | 0.183 | 0.243 | 0.157 | 0.252 | 0.387 | 0.276 |
| CD | CI | 96 | 0.146 | 0.195 | 0.165 | 0.229 | 0.125 | 0.222 | 0.362 | 0.266 |
| | | 192 | 0.188 | 0.239 | 0.178 | 0.245 | 0.145 | 0.241 | 0.374 | 0.274 |
| | | 336 | 0.242 | 0.284 | 0.191 | 0.252 | 0.166 | 0.263 | 0.388 | 0.282 |
| | | 720 | 0.310 | 0.331 | 0.201 | 0.261 | 0.205 | 0.293 | 0.423 | 0.296 |
| | | *Avg.* | 0.222 | 0.262 | 0.184 | 0.247 | 0.160 | 0.255 | 0.387 | 0.280 |
| CI | CD | 96 | 0.143 | 0.193 | 0.161 | 0.224 | 0.118 | 0.218 | 0.340 | 0.240 |
| | | 192 | 0.185 | 0.235 | 0.177 | 0.237 | 0.142 | 0.237 | 0.343 | 0.250 |
| | | 336 | 0.237 | 0.277 | 0.188 | 0.244 | 0.156 | 0.252 | 0.363 | 0.257 |
| | | 720 | 0.307 | 0.330 | 0.197 | 0.252 | 0.179 | 0.278 | 0.393 | 0.271 |
| | | *Avg.* | **0.218** | **0.259** | **0.181** | **0.239** | **0.149** | **0.246** | **0.360** | **0.255** |
| CD | CD | 96 | 0.146 | 0.197 | 0.162 | 0.229 | 0.125 | 0.221 | 0.352 | 0.254 |
| | | 192 | 0.188 | 0.238 | 0.178 | 0.245 | 0.148 | 0.246 | 0.361 | 0.266 |
| | | 336 | 0.241 | 0.282 | 0.191 | 0.254 | 0.161 | 0.261 | 0.377 | 0.267 |
| | | 720 | 0.313 | 0.333 | 0.199 | 0.260 | 0.189 | 0.289 | 0.412 | 0.288 |
| | | *Avg.* | 0.222 | 0.263 | 0.183 | 0.247 | 0.156 | 0.254 | 0.376 | 0.269 |

Table 15: Full results of ablation on modeling approaches for Integrated Attention and Cointegrated Attention. "CI" denotes channel independent and "CD" denotes channel-dependent modeling.

| Downsampled Patch Number M | Length | Weather MSE | Weather MAE | Solar MSE | Solar MAE | Electricity MSE | Electricity MAE | Traffic MSE | Traffic MAE |
|---|---|---|---|---|---|---|---|---|---|
| 1 | 96 | 0.171 | 0.231 | 0.172 | 0.234 | 0.135 | 0.231 | 0.349 | 0.252 |
|  | 192 | 0.207 | 0.261 | 0.187 | 0.252 | 0.158 | 0.255 | 0.358 | 0.259 |
|  | 336 | 0.257 | 0.298 | 0.196 | 0.257 | 0.182 | 0.278 | 0.382 | 0.269 |
|  | 720 | 0.323 | 0.344 | 0.203 | 0.258 | 0.191 | 0.290 | 0.414 | 0.282 |
|  | *Avg.* | 0.239 | 0.284 | 0.189 | 0.250 | 0.166 | 0.264 | 0.375 | 0.266 |
| 4 | 96 | 0.147 | 0.201 | 0.168 | 0.231 | 0.118 | 0.218 | 0.340 | 0.240 |
|  | 192 | 0.189 | 0.241 | 0.183 | 0.245 | 0.142 | 0.237 | 0.343 | 0.250 |
|  | 336 | 0.240 | 0.282 | 0.195 | 0.251 | 0.156 | 0.252 | 0.363 | 0.257 |
|  | 720 | 0.309 | 0.332 | 0.200 | 0.255 | 0.179 | 0.278 | 0.393 | 0.271 |
|  | *Avg.* | 0.221 | 0.264 | 0.186 | 0.246 | **0.149** | **0.246** | 0.360 | 0.255 |
| 8 | 96 | 0.144 | 0.195 | 0.163 | 0.225 | 0.119 | 0.219 | 0.338 | 0.240 |
|  | 192 | 0.186 | 0.237 | 0.183 | 0.243 | 0.146 | 0.244 | 0.341 | 0.249 |
|  | 336 | 0.238 | 0.280 | 0.195 | 0.251 | 0.161 | 0.260 | 0.379 | 0.264 |
|  | 720 | 0.308 | 0.330 | 0.196 | 0.250 | 0.177 | 0.277 | 0.400 | 0.280 |
|  | *Avg.* | 0.219 | 0.261 | 0.184 | 0.242 | 0.151 | 0.250 | 0.365 | 0.258 |
| 12 | 96 | 0.143 | 0.192 | 0.161 | 0.224 | 0.120 | 0.220 | 0.334 | 0.238 |
|  | 192 | 0.185 | 0.235 | 0.177 | 0.237 | 0.148 | 0.247 | 0.337 | 0.250 |
|  | 336 | 0.237 | 0.277 | 0.188 | 0.244 | 0.163 | 0.264 | 0.363 | 0.256 |
|  | 720 | 0.307 | 0.330 | 0.197 | 0.252 | 0.176 | 0.286 | 0.387 | 0.268 |
|  | *Avg.* | **0.218** | **0.259** | 0.181 | 0.239 | 0.151 | 0.254 | 0.355 | **0.253** |
| 16 | 96 | 0.145 | 0.195 | 0.149 | 0.223 | 0.122 | 0.223 | 0.333 | 0.235 |
|  | 192 | 0.187 | 0.238 | 0.175 | 0.236 | 0.149 | 0.249 | 0.343 | 0.254 |
|  | 336 | 0.241 | 0.282 | 0.187 | 0.244 | 0.165 | 0.265 | 0.354 | 0.256 |
|  | 720 | 0.312 | 0.328 | 0.196 | 0.250 | 0.179 | 0.276 | 0.386 | 0.269 |
|  | *Avg.* | 0.222 | 0.261 | **0.177** | **0.238** | 0.152 | 0.253 | **0.354** | 0.254 |

Table 16: Full results of varying the number of downsampled patches $M$ on forecasting performance.

| Model | TimeBridge MSE | TimeBridge MAE | PDF (2024) MSE | PDF (2024) MAE | Confidence Interval |
|---|---|---|---|---|---|
| ETTm1 | $0.353 \pm 0.014$ | $0.388 \pm 0.010$ | $0.339 \pm 0.008$ | $0.372 \pm 0.006$ | 99% |
| ETTm2 | $0.246 \pm 0.004$ | $0.310 \pm 0.012$ | $0.252 \pm 0.003$ | $0.313 \pm 0.003$ | 99% |
| ETTh1 | $0.399 \pm 0.010$ | $0.420 \pm 0.008$ | $0.399 \pm 0.015$ | $0.419 \pm 0.006$ | 99% |
| ETTh2 | $0.346 \pm 0.018$ | $0.394 \pm 0.015$ | $0.327 \pm 0.009$ | $0.376 \pm 0.010$ | 99% |
| Weather | $0.218 \pm 0.006$ | $0.259 \pm 0.004$ | $0.225 \pm 0.009$ | $0.261 \pm 0.006$ | 99% |
| Electricity | $0.149 \pm 0.011$ | $0.246 \pm 0.007$ | $0.159 \pm 0.010$ | $0.250 \pm 0.015$ | 99% |
| Traffic | $0.360 \pm 0.008$ | $0.255 \pm 0.013$ | $0.383 \pm 0.016$ | $0.254 \pm 0.010$ | 99% |
| Solar | $0.181 \pm 0.002$ | $0.239 \pm 0.003$ | $0.205 \pm 0.008$ | $0.265 \pm 0.005$ | 99% |

Table 17: Standard deviation and statistical tests for TimeBridge and second-best method (PDF) on ETT, Weather, Electricity, Traffic, and Solar datasets.

| Model | TimeBridge MAE | TimeBridge MAPE | TimeBridge RMSE | TimeMixer (2024) MAE | TimeMixer (2024) MAPE | TimeMixer (2024) RMSE | Confidence Interval |
|---|---|---|---|---|---|---|---|
| PeMS03 | $14.63 \pm 0.164$ | $14.21 \pm 0.133$ | $23.10 \pm 0.186$ | $14.63 \pm 0.112$ | $14.54 \pm 0.105$ | $23.28 \pm 0.128$ | 99% |
| PeMS04 | $19.24 \pm 0.131$ | $12.42 \pm 0.108$ | $31.12 \pm 0.112$ | $19.21 \pm 0.217$ | $12.53 \pm 0.154$ | $30.92 \pm 0.143$ | 99% |
| PeMS07 | $20.43 \pm 0.173$ | $8.42 \pm 0.155$ | $33.44 \pm 0.190$ | $20.57 \pm 0.158$ | $8.62 \pm 0.112$ | $33.59 \pm 0.273$ | 99% |
| PeMS08 | $14.98 \pm 0.278$ | $9.56 \pm 0.126$ | $23.77 \pm 0.142$ | $15.22 \pm 0.311$ | $9.67 \pm 0.101$ | $24.26 \pm 0.212$ | 99% |

Table 18: Standard deviation and statistical tests for TimeBridge and second-best method (TimeMixer) on the PeMS dataset.

| Model | TimeBridge | | | | | | Crossformer (2023) | | | | | | Confidence |
|---|---|---|---|---|---|---|---|---|---|---|---|---|---|
| Dataset | ARR | AVol | MDD | ASR | CR | IR | ARR | AVol | MDD | ASR | CR | IR | Interval |
| CSI 500 | 0.285 ± 0.033 | 0.203 ± 0.012 | −0.196 ± 0.010 | 1.405 ± 0.016 | 1.453 ± 0.021 | 1.317 ± 0.019 | −0.039 ± 0.027 | 0.163 ± 0.009 | −0.217 ± 0.011 | −0.238 ± 0.021 | −0.179 ± 0.014 | −0.350 ± 0.014 | 95% |
| S&P 500 | 0.326 ± 0.022 | 0.169 ± 0.009 | −0.142 ± 0.010 | 1.927 ± 0.017 | 2.298 ± 0.019 | 1.842 ± 0.016 | 0.284 ± 0.024 | 0.159 ± 0.011 | −0.114 ± 0.014 | 1.786 ± 0.014 | 2.491 ± 0.018 | 1.842 ± 0.012 | 95% |

Table 19: Standard deviation and statistical tests for TimeBridge and second-best method (Crossformer) on the CSI 500 and S&P 500 dataset.

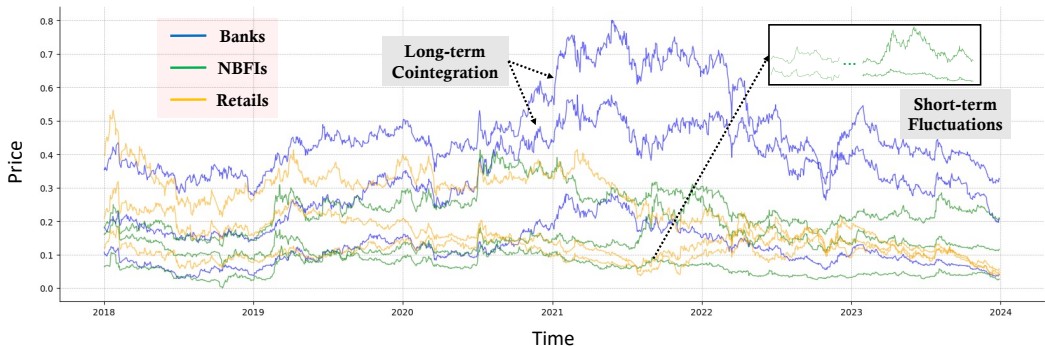

Figure 7: Visualization of short-term fluctuations and long-term cointegration across stock sectors. "NBFIs" represents Non-Bank Financial Institutions. The figure highlights how sectors experience short-term price volatility while maintaining long-term cointegration.

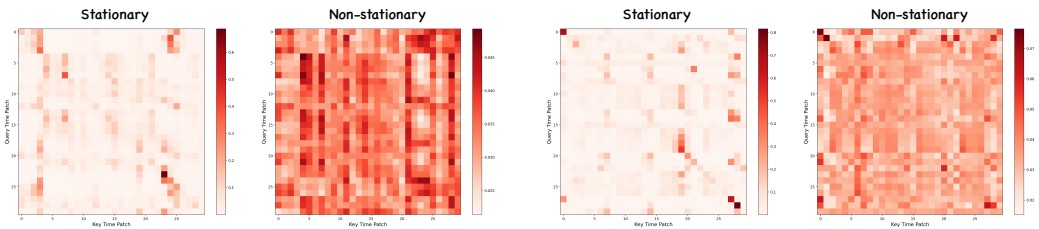

Figure 8: Additional examples comparing intra-variate attention maps under stationary and non-stationary conditions for different patches in the Electricity dataset.

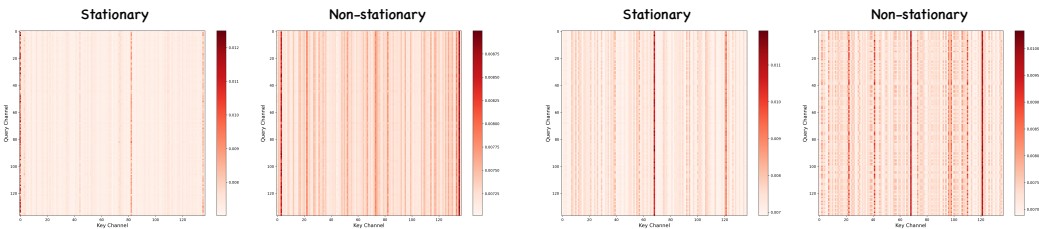

Figure 9: Additional examples comparing inter-variate attention maps between different variates under stationary and non-stationary conditions in the Solar dataset.

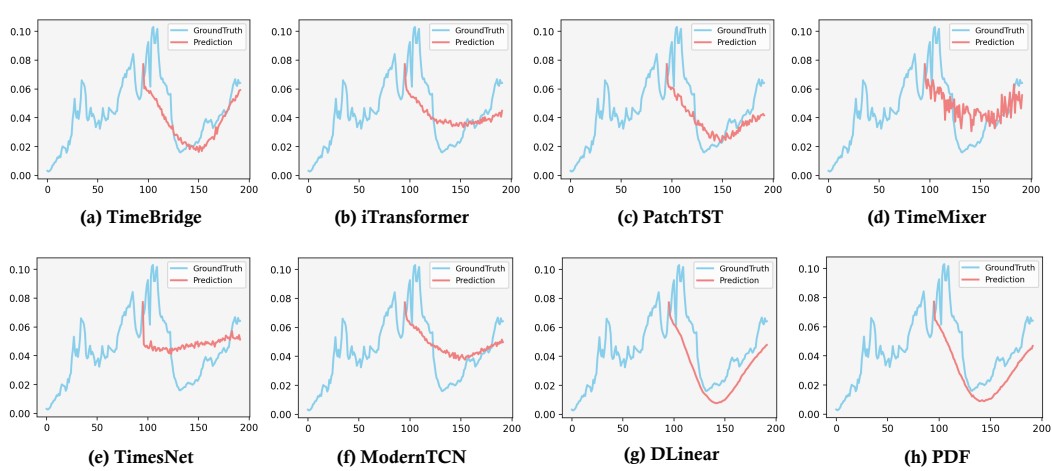

Figure 10: Visualization of predictions from different models on the Weather dataset.

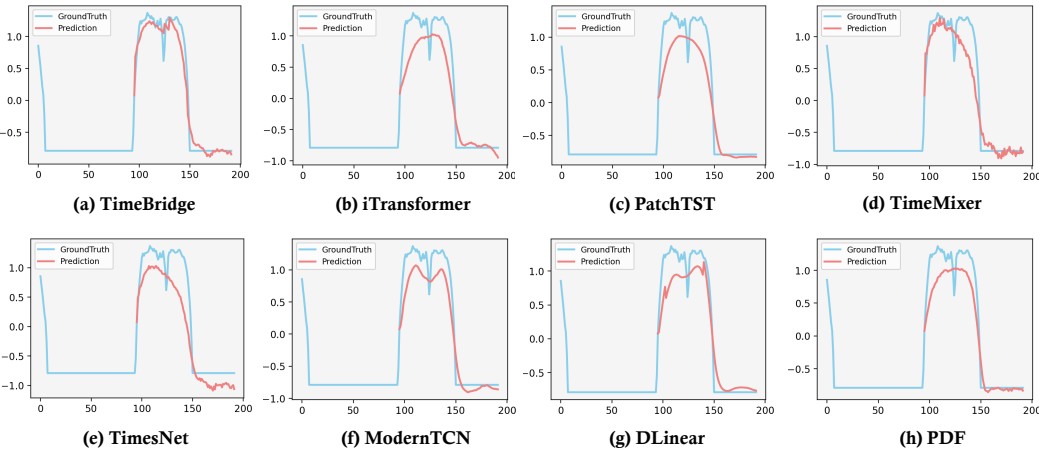

Figure 11: Visualization of predictions from different models on the Solar dataset.

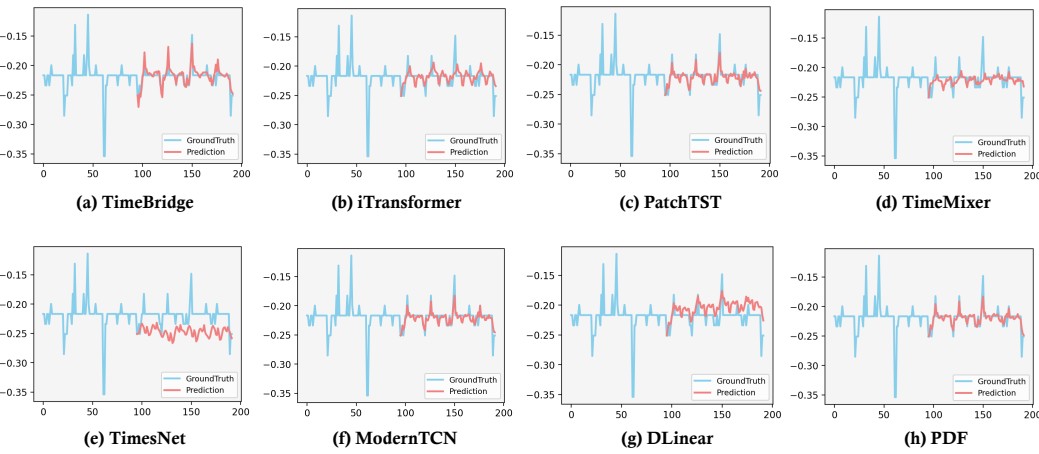

Figure 12: Visualization of predictions from different models on the Electricity dataset.

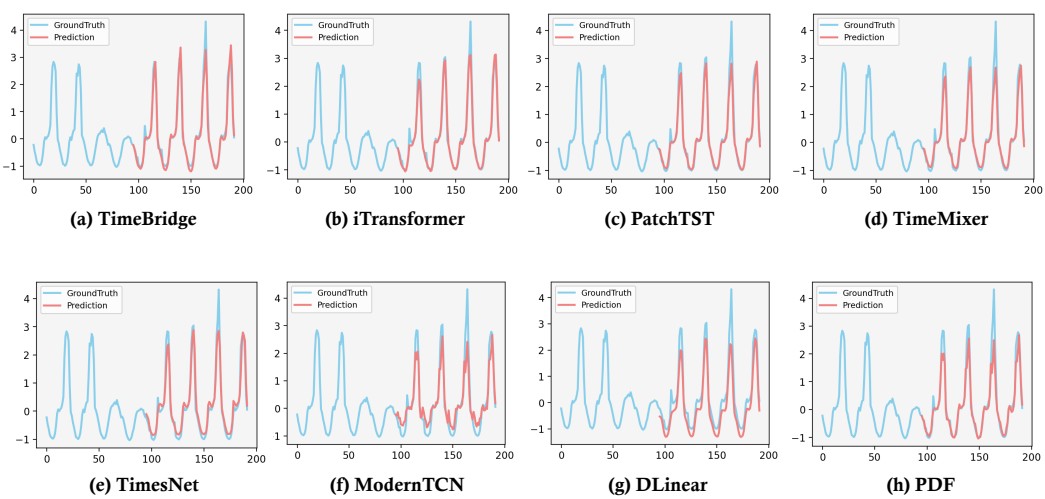

Figure 13: Visualization of predictions from different models on the Traffic dataset.

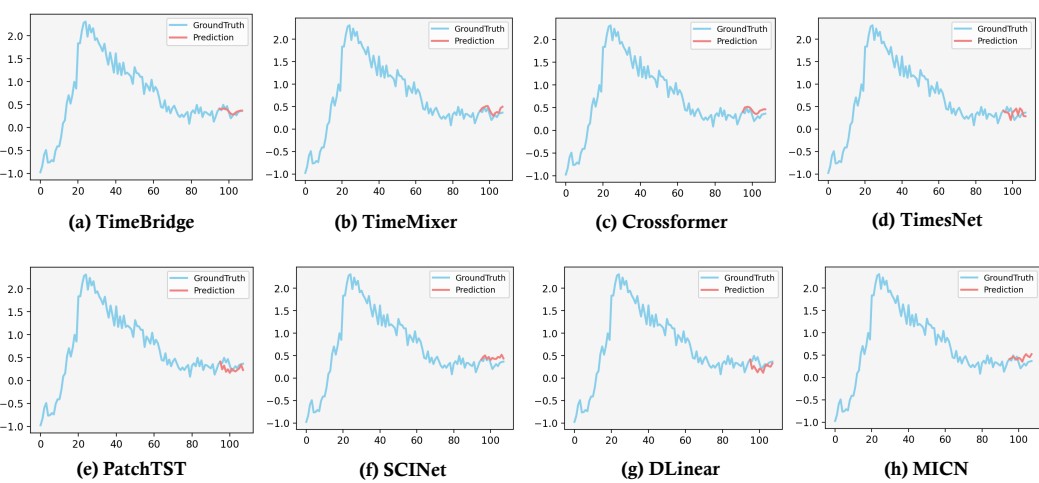

Figure 14: Visualization of predictions from different models on the PeMS03 dataset.

