# OpenReview forum: "TimeBridge: Non-Stationarity Matters for Long-term Time Series Forecasting"
_ICLR.cc/2025/Conference — Submitted to ICLR 2025_

### Official Review · Reviewer_nS1F · 2024-11-02

**Soundness:** 2
**Presentation:** 2
**Contribution:** 2
**Rating:** 3
**Confidence:** 4

**Summary:**

This paper introduces TimeBridge for predicting non-stationary time series, aiming to resolve the contradiction between short-term fluctuations and long-term trends. By dividing the input sequence into small segments and applying an integrated attention mechanism to reduce short-term instability while utilizing joint attention mechanisms to preserve non-stability to capture long-term co-integration relationships across variables, TimeBridge effectively captures stable dependencies without introducing pseudo-regression risks. Experimental results demonstrate that TimeBridge performs well on multiple tasks, particularly outperforming current state-of-the-art methods in financial time series prediction. Additionally, through a series of ablation studies, the importance of removing or preserving non-stability and the order of integrated and joint attention modules is further validated, revealing optimal model configurations under different dataset backgrounds. This work not only enhances the accuracy and robustness of time series prediction but also provides new perspectives and methods for handling non-stationary data in the future.

**Strengths:**

- Resolving the conflict between short-term fluctuations and long-term trends: TimeBridge can effectively capture stable dependencies by reducing short-term non-stability through dividing the input sequence into small fragments and applying integrated attention mechanisms, while utilizing joint attention mechanisms to preserve non-stability to capture cross-variable long-term co-integration relationships.

- Improving the accuracy and robustness of time and sequence prediction: TimeBridge performs well on multiple tasks, particularly outperforming current state-of-the-art methods in financial time series prediction.

- Providing new perspectives and methods for handling non-stationary data in the future: Through a series of ablation studies, this article further verifies the importance of removing or retaining non-stability, as well as the sequence of integrated and joint attention modules, and reveals the optimal model configuration under different datasets, providing new perspectives and methods for handling non-stationary data in the future.

**Weaknesses:**

- The paper lacks sufficient originality, as it primarily builds upon existing methods without presenting a clear, novel contribution to the field. This limits the manuscript's potential impact and reduces its value as an advancement in research.

- The lack of experiments with varying input/output (I/O) ratios raises concerns about the fairness of the comparisons. Different I/O settings can affect model outcomes substantially, and without exploring these variations, the paper provides an incomplete evaluation of model performance.

- The authors have only considered MSE and MAE as metrics for their long-term forecasting task, which is insufficient. This limited evaluation may obscure important aspects of model accuracy, particularly in cases where relative error size is a crucial factor for assessing model effectiveness.

- The paper lacks a clear description of the hyperparameter tuning process for the comparative algorithms, which raises concerns about the comparisons. Without adequate tuning, it is difficult to ascertain if the reported performance differences are truly reflective of each model's capabilities.

**Questions:**

- The experimental settings are not reasonable. Why input length is set to 720, output length O is set to 96, 192, 336, 720 and not something more practical, like 1 month of data?

- The paper does not account for the impact of varying input/output (I/O) ratios, which can significantly influence model performance. Fixing different input length for baselines without comprehensive experiments may lead to biased comparisons and limit the robustness of the results. The author needs to report the results of the baseline under the same experimental setup.

- The paper lacks an evaluation of the algorithm's complexity, particularly with respect to its theoretical complexity, GPU resource cost, and runtime efficiency. Without this assessment, it is difficult to gauge the algorithm's practicality and scalability, especially in resource-constrained environments.

- The absence of detailed hyperparameter optimization for the baseline algorithms undermines the fairness of the comparisons presented in the paper. Without this tuning, the results may not accurately represent the optimal performance of the comparative models, potentially leading to biased conclusions.

---

> ### Author Response · Authors · 2024-11-14
>
> Thank you for your feedback and valuable comments. We greatly appreciate the time you spent reviewing our paper. Here are detailed responses to your concerns and questions.
>
> **Q1:** The paper lacks originality, primarily building on existing methods without presenting a clear, novel contribution.
>
> **A1:** We propose a novel approach **by integrating the modeling of non-stationarity and dependency in long-term time series forecasting—an area where previous works typically address only one of these aspects**. In particular, **Integrated Attention** is innovative in how it addresses non-stationarity while simultaneously capturing short-term dependencies within individual variables, achieved by manipulating the attention map directly. **Cointegrated Attention** is the first application of the concept of cointegration to long-term time series forecasting. Cointegration is a well-established statistical method in econometrics but has not been applied to forecasting, making its use in this context a novel contribution. We appreciate Reviewer 5kbC's recognition of this as an important advancement in time series forecasting.
>
> ---
> **Q2:** The paper doesn’t explore varying input/output (I/O) ratios, which may impact model performance and fairness in comparisons.
>
> **A2:** We have addressed this concern in our experiments. In **Table 2** of the main paper, and the extended results in **Table 11**, we perform an exhaustive search for the optimal input lengths for all baselines. Specifically, we tested input lengths $I \in 96,192,336,512,720$, and the full search process is detailed in **Appendix E.1 (lines 903-908)**.
>
> ---
> **Q3:** The evaluation only considers MSE and MAE, which may not fully capture model effectiveness, especially for relative error.
>
> **A3:** While MSE and MAE are the standard evaluation metrics used in the time series forecasting literature, including many influential papers such as TimeMixer, iTransformer, PatchTST, and TimesNet, we understand your concern. In response, we have added additional evaluation metrics for relative error: MAPE and MSPE, as shown in **Table A** in the following comment. These additional metrics further demonstrate the effectiveness of our approach across different evaluation criteria.
>
> ---
> **Q4:** The paper does not clearly describe the hyperparameter tuning process for the baseline models, raising concerns about fairness in the comparisons.
>
> **A4:** We apologize for any confusion. As mentioned in **Answer 2**, we have performed extensive hyperparameter tuning for all baseline models. The results reported in Table 2 (full results in Table 11) are based on the best-performing configurations for each model, after tuning key hyperparameters such as learning rates, encoder layers, model dimensions, and training epochs. These details are fully described in **lines 906-907**.
>
> ---
> **Q5:** The experimental settings are not reasonable. Why is the input length set to 720, output length $O$ set to 96,192,336,720, and not something more practical, like 1 month of data?
>
> **A5:** The input length $I$ was selected was selected through a search over $I \in 96,192,336,512,720$, with the best-performing length chosen. For output lengths, $O \in 96,192,336,720$ is commonly used in time series forecasting literature. For example, in Informer [1], for datasets like **ETT**, an output length of 96 corresponds to 1 month for ETT, where the time interval is 15 minutes (15 mins * 96 = 1 month). For **Electricity** and **Traffic** (1-hour interval), 720 also represents 1 month. This output configuration is both practical and widely adopted.
>
> ---
> **Q6:** The paper does not assess the algorithm’s complexity, particularly in terms of computational cost, GPU usage, and runtime efficiency.
>
> **A6:** We include a detailed comparison of the theoretical complexity of our method and other Transformer-based models. The table below summarizes the theoretical computational complexity, where $C$ is the number of channels, $I$ and $O$ are the input and output lengths, and $p$ is the patch length:
> |TimeBridge|iTransformer|PatchTST|Crossformer|FEDformer|
> |-|-|-|-|-|
> |$O(\text{max}(C·(\frac{I}{p})^2，C^{3/2}))$|$O(C^2)$|$O(C·(\frac{I}{p})^2)$|$O(\frac{C}{p^2}O(I+O))$|$O(I/2+O)$|
>
> Moreover, theoretical complexity alone cannot fully capture real-world performance due to implementation differences. We tested on 2 NVIDIA 3090 GPUs, measuring training (1 epoch) and inference times for three datasets of increasing size, with results averaged over 5 runs in **Table B** in the following comment.
>
> ---
> Overall, thank you for your valuable comments. We hope the detailed experiments and clarifications provided above address your concerns. If you have any further questions, please feel free to reach out, and we would be happy to provide additional clarifications. We kindly ask you to reconsider your evaluation in light of these explanations.
>
> [1] Informer: Beyond Efficient Transformer for Long Sequence Time-Series Forecasting

---

> > ### Comment · Reviewer_nS1F · 2024-11-22
> >
> > Thanks for the response. While the authors have addressed some concerns and provided additional data, the innovative contribution is still found lacking. There are ongoing concerns about the consistency of the results across different computational environments, suggesting potential variability in outcomes. Furthermore, the performance of the algorithm does not significantly outperform PDF, which is a key consideration for acceptance at ICLR where methodological innovation is highly valued. The contribution to advancing the field is therefore deemed insufficient, leading to the maintenance of the initial score.

---

> ### Author Response · Authors · 2024-11-14
> **Additional Experimental Results**
>
> ### **Table A:** Additional MAPE (Mean Absolute Percentage Error) and MSPE (Mean Squared Percentage Error) scores, both of which measure relative prediction error.
> |||TimeBridge|PDF|ModernTCN|TimeMixer|
> |-|-|-|-|-|-|
> |||MAPE/MSPE|MAPE/MSPE|MAPE/MSPE|MAPE/MSPE|
> |ETTh2|96|**1.38**/**262.4**|1.41/344.6|1.47/378.3|1.53/395.1|
> ||192|**1.56/331.2**|1.56/366.4|1.59/376.9|1.66/442.9|
> ||336|1.81/472.5|**1.701/435.0**|1.74/495.4|1.85/506.5|
> ||720|2.30/833.5|2.09/667.5|**2.00/579.1**|2.35/798.4|
> |Weather|96|**10.18/1.08e7**|11.974/1.44e7|10.30/1.15e7|10.63/1.13e7|
> ||192|**12.72/1.25e7**|13.648/1.88e7|12.92/1.46e7|13.65/1.97e7|
> ||336|12.97/1.87e7|14.040/2.26e7|**9.62/1.21e7**|12.00/1.59e7|
> ||720|**13.78/2.01e7**|15.147/2.45e7|14.32/2.09e7|15.64/2.66e7|
> |Electricity|96|**2.17/2.89e5**|2.34/6.86e5|2.57/1.04e6|2.29/3.69e5|
> ||192|**2.28/3.46e5**|2.99/4.08e6|2.84/1.11e6|2.56/3.95e5|
> ||336|**2.43/3.33e5**|3.18/5.86e5|2.89/8.10e5|2.75/4.82e5|
> ||720|**2.65/2.84e5**|3.08/3.67e5|2.88/4.01e5|2.73/4.35e5|
> |Solar|96|**1.62/1.08e4**|1.78/2.10e4|1.86/2.77e4|2.04/2.89e4|
> ||192|**2.02/3.44e4**|2.05/3.60e4|2.29/4.02e4|2.35/4.66e4|
> ||336|**2.12/3.72e4**|2.20/3.85e4|2.40/4.59e4|2.46/4.75e4|
> ||720|**2.15/3.78e4**|2.26/3.91e4|2.43/6.21e4|2.48/4.76e4|
>
> ---
> ---
>
> ### **Table B:** Running time of different Transformer-based methods.
> |||TimeBridge|iTransformer|PatchTST|Crossformer|FEDformer|
> |-|-|-|-|-|-|-|
> |ETTm1 ($C=7$)|Training Time|72s|65s|73s|280s|449s|
> ||Inference Time|33s|31s|39s|48s|62s|
> |Electricity ($C=321$)|Training Time|252s|89s|450s|328s|510s|
> ||Inference Time|125s|78s|141s|116s|130s|
> |Traffic ($C=862$)|Training Time|409s|175s|649s|360s|485s|
> ||Inference Time|175s|154s|207s|171s|196s|

---

> ### Author Response · Authors · 2024-11-17
> **Follow-Up on Rebuttal Discussion**
>
> Dear Reviewer nS1F,
>
> Thank you for your valuable feedback during the first round of review. Your comments have provided us with significant insights, and we have conducted detailed experiments to address your concerns thoroughly. We believe these new results offer additional clarity and reinforce the contributions of our work.
>
> We kindly hope you might reconsider your concerns in light of our response and the additional experiments. We would greatly appreciate any further feedback or suggestions you might have, as they would be invaluable in helping us improve our work. Looking forward to your reply.
>
> Best regards,
>
> Paper3735 Authors

---

### Official Review · Reviewer_aC7S · 2024-11-04

**Soundness:** 2
**Presentation:** 3
**Contribution:** 3
**Rating:** 3
**Confidence:** 4

**Summary:**

The paper presents TimeBridge for non-stationary time series forecasting. TimeBridge utilizes Integrated Attention to mitigate short-term non-stationarity and Cointegration Attention for modeling long-term non-stationarity. Experiment results on benchmarking time series demonstrate the effectiveness of TimeBridge.

**Strengths:**

+ Non-stationarity is a major challenge in time series forecasting. The paper aims to address this important task
+ Experiment results are encouraging, with a comprehensive ablation study.

**Weaknesses:**

- The idea of Integrated Attention and Cointegration Attention is not new. The model in general lacks novelty.

- The improvement by TimeBridge is not as much as claimed in the paper (over 10%). On most of the datasets, TimeBridge achieves similar results or marginally better results (Table 1 and 2). In general, the popular benchmarking time series are relatively easier tasks. On finance applications, TimeBridge's performance is similar to TSmixer.

**Questions:**

What is TimeBridge's performance on bigger time series dataset, such as New York Taxi or Climate Data?

---

> ### Author Response · Authors · 2024-11-14
>
> Thank you for your feedback and insightful comments on our work. We appreciate the time you took to review our manuscript. Below, we address each of your concerns and questions:
>
> **Q1:** The idea of Integrated Attention and Cointegration Attention is not new. The model, in general, lacks novelty.
>
> **A1:** While it is true that **traditional time series forecasting** has explored related concepts, our work is **the first to bridge the gap between non-stationarity and dependency modeling in the context of long-term forecasting using deep learning**. We provide a comprehensive review of prior works, which primarily focus on either non-stationarity or dependency, but fail to address the interaction between these two aspects. The proposed **Integrated Attention** is innovative in its approach to removing non-stationarity while capturing short-term dependencies within variables by directly manipulating the attention map, which is a novel aspect of our model. Furthermore, the **Cointegrated Attention** introduces the concept of cointegration into long-term time series forecasting, marking the first time this technique has been applied in this context. We are encouraged by Reviewer 5kbC's recognition of this contribution as a novel step in the field.
>
> ---
> **Q2:** The improvement by TimeBridge is not as much as claimed in the paper (over 10%). On most of the datasets, TimeBridge achieves similar results or marginally better results (Table 1 and 2). The popular benchmarking time series are relatively easier tasks.
>
> **A2:** We apologize for any confusion caused. The **over 10% improvement** mentioned in the paper refers to the comparison with the baseline results reported in **Table 1**, where we compute improvements based on **the top two methods** (i.e., PDF and PatchTST) across all datasets. As for Table 2, the results include hyperparameter tuning for all baselines. In the paper, we specifically detail TimeBridge's performance compared to other state-of-the-art methods, such as Transformer-based PDF, CNN-based ModernTCN, and MLP-based TimeMixer. For example, TimeBridge achieves a 3.10%/1.64% reduction in MSE/MAE compared to PDF, 3.55%/0.81% compared to ModernTCN, and 6.92%/4.54% compared to TimeMixer. We hope this clarifies the magnitude of improvement. While the benchmark datasets may seem easier, they are widely adopted in time series forecasting research, and they encompass a variety of volatility and non-stationarity patterns. **Moreover, we have conducted experiments on larger datasets**, as discussed in Q4.
>
> ---
> **Q3:** On finance applications, TimeBridge's performance is similar to TSMixer.
>
> **A3:** We believe there might have been a slight misunderstanding. In **Table 4**, the model you referred to as **TSMixer** is intended to be **TimeMixer**, which was the correct baseline used in our study. To ensure we fully address your concern, we have now added the results for TSmixer [1] (which is a different baseline method) in **Table A** in following comment. When comparing TimeBridge with TSMixer and other deep learning-based methods, our results show that TimeBridge outperforms the other methods on most metrics, achieving the best performance in 7 out of 12 metrics. The second-best method, Crossformer, only outperforms TimeBridge in 3 out of 12 metrics. We kindly ask you to reconsider our model’s performance in finance applications based on this new information.
>
> ---
> **Q4:** What is TimeBridge's performance on bigger time series datasets, such as New York Taxi or Climate Data?
>
> **A4:** In this work, we have evaluated larger-scale datasets, including the **Traffic dataset (862 channels)** and **PEMS 07 (883 channels)**, which are considered challenging in prior studies. For example, the Traffic dataset includes hourly road occupancy data from 862 detectors across the San Francisco Bay Area (2015-2016), **totaling over 1,000,000 data points**. Additionally, we conducted experiments on a larger climate dataset, which records regional precipitation data across 1,763 stations in Australia from [2]. This climate dataset is characterized by **high-dimensionality (1,763 channels) and a long time span, with each channel containing 10828-time points**. The results, where TimeBridge outperforms all other baselines, are presented in **Table B** in the following comment.
>
> ---
> We sincerely appreciate your constructive comments. If our responses have addressed your concerns, we kindly ask if you could reconsider your evaluation. If you have any further questions or need additional clarification, please feel free to reach out, and we would be happy to assist.
>
> [1] TSMixer: An All-MLP Architecture for Time Series Forecasting
>
> [2] Monash Time Series Forecasting Archive

---

> > ### Comment · Reviewer_aC7S · 2024-11-22
> >
> > I appreciate the authors' efforts to address my concerns. But I'm afraid that the additional experiments do not change the overall assessment that there is limited improvement of TimeBridge compared with state-of-art methods. The confidence interval has not been reported, which typically puts most methods, such as transformer based approaches, DLinear, etc, in similar range. This is also aligned with the novelty concern. Slightly modifying the architecture with various attentions will not lead to drastic improvement of the performance. I therefore will maintain my score.

---

> > > ### Author Response · Authors · 2024-11-22
> > > **Response to Reviewer aC7S**
> > >
> > > Thank you for your reply and for taking the time to review our work. We would like to clarify two key points regarding the novelty and results of our work:
> > >
> > > 1. Novelty: **We believe that overly complex designs are unnecessary.** As demonstrated by iTransformer [1] (ICLR 2024 spotlight paper), their work achieved impressive results by "slightly" modifying self-attention by inverting it to better align with temporal data. **Similarly, our focus is to address a fundamental problem in time series forecasting—non-stationarity and cointegration—which has not been explored before in time series literature.** Our design is intentionally simple, aiming to solve this problem efficiently while ensuring the architecture remains effective. This focus on addressing a previously unaddressed issue in a concise and practical way is, in our view, the core of our novelty.
> > >
> > > 2. Results: As shown in Table 2-3 and the additional experiments in Table B, TimeBridge demonstrates improvements over state-of-the-art methods. In challenging multivariate forecasting tasks with many channels (e.g., Solar, Electricity, Traffic), our method achieves significant improvements, as evidenced by the fair hyperparameter tuning comparisons in Table 2. Improvements on other datasets are also acceptable. **Regarding confidence intervals, we have already reported them in Appendix Tables 17, 18, and 19, where TimeBridge consistently outperforms the second-best methods across three different tasks.**
> > >
> > > We hope these clarifications address your concerns and provide further context to the contributions of our work.
> > >
> > >
> > > [1] iTransformer: Inverted Transformers Are Effective for Time Series Forecasting. ICLR 2024.

---

> ### Author Response · Authors · 2024-11-14
> **Additional Experimental Results**
>
> ### **Table A:** Additional baselines (TSMixer) results for financial forecasting.
> ||ARR$\uparrow$|AVol$\downarrow$|MDD$\downarrow$|ASR$\uparrow$|CR$\uparrow$|IR$\uparrow$|ARR$\uparrow$|AVol$\downarrow$|MDD$\downarrow$|ASR$\uparrow$|CR$\uparrow$|IR$\uparrow$|
> |-|-|-|-|-|-|-|-|-|-|-|-|-|
> ||CSI500||||||S&P500||||||
> |PatchTST|0.118|**0.152**|-0.127|0.776|0.923|0.735|0.146|0.167|-0.140|0.877|1.042|0.949|
> |Crossforme|-0.039|0.163|-0.217|-0.238|-0.179|-0.350|0.284|**0.159**|**-0.114**|1.786|**2.491**|1.646|
> |iTransformer|0.214|0.168|-0.164|1.276|1.309|1.173|0.159|0.170|-0.139|0.941|1.150|0.955|
> |TimeMixer|0.078|0.153|**-0.114**|0.511|0.685|0.385|0.254|0.162|-0.131|1.568|1.938|1.448|
> |TSMixer (Add)|0.086|0.156|-0.143|0.551|0.601|0.456|0.187|0.173|-0.156|1.081|1.199|1.188|
> |TimeBridge (Ours)|**0.285**|0.203|-0.196|**1.405**|**1.453**|**1.317**|**0.326**|0.169|-0.142|**1.927**|2.298|**1.842**|
>
> ---
> ---
>
> ### **Table B:** Additional Results of larger climate dataset.
> ||TimeBridge|iTransformer|PDF|TimeMixer|PatchTST|Crossformer|FEDformer|ModernTCN|MICN|TimesNet|DLinear|
> |-|-|-|-|-|-|-|-|-|-|-|-|
> ||MSE/MAE|MSE/MAE|MSE/MAE|MSE/MAE|MSE/MAE|MSE/MAE|MSE/MAE|MSE/MAE|MSE/MAE|MSE/MAE|MSE/MAE|
> |96 |**1.040**/**0.497**|1.788/0.816|1.116/0.530|1.341/0.578|1.354/0.519|1.083/0.508|1.290/0.620|1.100/0.514|1.150/0.574|1.274/0.524|1.108/0.536|
> |192|**1.049**/**0.494**|1.837/0.828|1.362/0.570|1.447/0.622|1.388/0.572|1.115/0.532|1.447/0.693|1.125/0.523|1.159/0.568|1.274/0.541|1.122/0.536|
> |336|**1.066**/**0.495**|1.931/0.869|1.396/0.568|1.430/0.610|1.410/0.581|1.246/0.575|1.293/0.625|1.153/0.530|1.173/0.571|1.285/0.584|1.146/0.545|
> |720|**1.071**/**0.496**|2.199/0.957|1.386/0.552|1.438/0.615|1.395/0.575|1.236/0.584|1.480/0.710|1.243/0.599|1.183/0.568|1.347/0.626|1.158/0.545|

---

> ### Author Response · Authors · 2024-11-17
> **Follow-Up on Rebuttal Discussion**
>
> Dear Reviewer aC7S,
>
> We deeply appreciate your valuable feedback during the first round of review and the thoughtful discussion that has significantly helped us refine our work. We hope that our responses have addressed your concerns.
>
> We sincerely appreciate the opportunity to refine our work and kindly hope you might reconsider your concerns in light of our detailed responses and clarifications. Your insights are invaluable to us, and we would greatly welcome any additional feedback or suggestions you may have. Thank you once again for your time and thoughtful consideration. Looking forward to your reply.
>
> Best Regards,
>
> Paper3735 Authors

---

### Official Review · Reviewer_5KbC · 2024-11-04

**Soundness:** 3
**Presentation:** 3
**Contribution:** 4
**Rating:** 8
**Confidence:** 4

**Summary:**

The paper introduces TimeBridge, a framework for multivariate time series forecasting that addresses the challenge of non-stationarity by differentiating between its impacts on short-term and long-term modeling. TimeBridge utilizes Integrated Attention to manage short-term fluctuations in batches, reducing spurious regressions and capturing local dependencies. Cointegrated Attention is introduced to preserve long-term non-stationarity across variates, enabling effective long-term dependency capture. Experiments on the CSI 500 and S&P 500 indices verify the short- and long-term forecasting performance. It is generally a method for handling nuanced non-stationarity effects in complex multivariate scenarios.

**Strengths:**

1. This paper is well written. The notations are clear.

2. It provides up-to-date literature on learning techniques for capturing stationary/non-stationary and dependency in sequential data. It combines the treatments of non-stationarity and dependency modeling in one shoot and delivers convincing performances.

3. The notion of cointegration of time series has been missing or forgotten in recent time series forecasting literature; this paper showed how cointegration could help reveal the non-stationary part when multiple time series evolve simultaneously.
In Operations Research, cointegration is a well-established technique, and it has been introduced to machine learning literature for long:

Marco Cuturi, Alexandre D’Aspremont (ICML, 2013). https://proceedings.mlr.press/v28/cuturi13.html

The authors may refer to basic cointegration techniques to resolve the computational challenges or benchmark the extraction of stationary/non-stationary movements.

4. The experiments are comprehensive and cover recent state-of-the-art competing methods. The results are convincing, as shown by a solid ablation study showing the contribution of the building blocks, e.g., Integrated Attention and Cointegrated Attention.

**Weaknesses:**

1. According to Figure 3, the building blocks of the proposed methods are streamlined with no conjugation. In some sense, this is brute force and remains room for improvement or further technical development, speaking of the systematic organic treatment of the non-stationarity and dependency modeling in long-term time-series forecasting.

**Questions:**

Q. As mentioned in Strength 3, it would be interesting to recap traditional techniques for cointegration in operations research and benchmark the stationary or non-stationary time series before discussing the impact of an attention-powered module.

---

> ### Author Response · Authors · 2024-11-14
>
> We sincerely appreciate your positive feedback and valuable insights into our work. Thank you for recognizing our effort in bringing the concept of cointegration into long-term time series forecasting, as well as for acknowledging the clarity of our notations, the novelty of our approach, and the comprehensiveness of our experiments. Below, we address your concerns and respond to your thoughtful questions in detail:
>
> **Q1:** Figure 3 suggests that the proposed method's building blocks are streamlined and lack integration, leaving room for more systematic treatment of non-stationarity and dependency modeling.
>
> **A1:** We would like to clarify that our method has been carefully designed with a simple and intuitive network structure, systematically addressing the challenges of non-stationarity and dependency modeling. Specifically:
> - **Integrated Attention** is tailored to capture short-term dependencies within variables while mitigating spurious regression. It achieves this by modeling non-stationarity in a streamlined yet effective manner, utilizing a simplified and innovative technique (i.e., the normalized attention map) to eliminate and reconstruct non-stationary components.
> - **Cointegrated Attention** is designed to preserve non-stationarity and model the long-term cointegration relationships across variables.
> - **Patch Downsampling** serves as a bridge between the two blocks, enabling Integrated Attention Block to process aggregated short-term information that benefits the downstream Cointegrated Attention Block.
>
> To validate our architectural decisions, we conducted an ablation study (Table 6 and Figure 6), which demonstrated the individual contributions of the Integrated and Cointegrated Attention Blocks.
> **The choice of a serial architecture not only proves highly effective in capturing both short-term dependencies and long-term cointegrations but also makes the model easy to understand.** Furthermore, the results highlight the necessity of modeling short-term dependencies before long-term ones. This ordering ensures that aggregated short-term information enhances subsequent long-term modeling, while reversing the order leads to information loss and performance degradation.
>
> ---
> **Q2:** It would be helpful to revisit traditional cointegration techniques and benchmark stationary and non-stationary series before evaluating the attention-powered module.
>
> **A2:** Thank you for this valuable suggestion. We will incorporate additional background on traditional cointegration techniques in operations research and other potential domains in the revised version of our manuscript. Specifically, we will expand the Normalization and Dependency Modeling sections in the Related Work to provide a more comprehensive overview of cointegration methods and their relevance. This will serve as a foundation for contextualizing the impact of our proposed attention-powered modules.
>
> ---
> Once again, we are deeply grateful for your recognition of our work and your constructive feedback. If you have any further questions or suggestions, we would be more than happy to address them.

---

> ### Author Response · Authors · 2024-11-17
> **Follow-Up on Rebuttal Discussion**
>
> Dear Reviewer 5KbC,
>
> Thank you for your positive feedback and high evaluation of our work. We are truly grateful for your recognition and would be delighted to hear any additional feedback or suggestions you may have to further improve the study.
>
> Thank you again for your time and support. Looking forward to your reply.
>
> Best regards,
>
> Paper3735 Authors

---

> > ### Comment · Reviewer_5KbC · 2024-11-27
> >
> > I thank the authors for their responses. Most of my concerns are addressed reasonably. Therefore, I remain my score on this paper.

---

> > > ### Author Response · Authors · 2024-11-28
> > >
> > > Thank you for your constructive review and valuable feedback throughout the reviewing process. We greatly appreciate your recognition and continued support of our work.

---

### Meta-Review · Area_Chair_Qmdu · 2024-12-17

**Metareview:**

**(a) Summary**

This paper addresses the task of long-term time series prediction by tackling the challenge of incorporating both short-term and long-term dependencies. The proposed method, called TimeBridge, combines two types of attention mechanisms: integrated attention on patches of time series for short-term modeling, followed by cointegrated attention to capture long-term dependencies. The empirical effectiveness of TimeBridge has been evaluated on multiple real-world datasets.

**(b) Strengths**

- **Relevance and motivation:** The paper is well-motivated, as the challenge of effectively modeling both short-term and long-term dependencies, particularly considering the non-stationarity of time series, is a critical problem in time series prediction.
- **Architecture design:** While the individual components of the proposed architecture are not entirely novel, their combination into the overall framework is unique and can be a valuable contribution to the community.

**(c) Weaknesses**

- **Hyperparameter tuning**: Hyperparameter optimization lacks sufficient analysis and discussion despite its importance to empirical performance. The explanation provided (e.g., Lines 905–906) is insufficient. Given the inherent challenges of hyperparameter tuning in time series prediction, a more thorough exploration of the process and sensitivity analysis with respect to key parameters is essential.
- **Datasets and experimental settings:** There is a notable error, which is the claim that 15 minutes * 96 equals one month for the Weather dataset, while in fact this corresponds to one day. This issue was raised during the AC-reviewer discussion phase and highlights the need for more rigorous verification of the dataset domains and associated details.
- **Empirical performance:** The empirical results are not convincing compared to existing approaches, as noted by multiple reviewers. Further evidence is needed to justify the superiority of the proposed method. Please also see my comment on (d).
- **Presentation:** There are several issues in paper’s readability and overall quality, which I strongly recommend to address throughout the paper. For example, "Patching(X)" is defined in Equation (1) but never used in the paper, and the matrix $\mathbf{P}'\_{c,:}$ is defined as $\\{\mathbf{p}\_1', \dots, \mathbf{p}\_N'\\}$ in Equation (2) but it means a set, where duplication is not allowed by definition and the size can be smaller than $N$, hence it is not appropriate to use it to define a matrix. In Equation (3), the function $\text{Attention}(\mathbf{P}\_{c,:}', \mathbf{P}\_{c,:}', \mathbf{P}\_{c,:})$ is not mathematically defined in the paper (although it can be inferred). While each issue may seem minor, collectively they significantly impact the paper’s clarity and quality.


**(d) Decision Reasoning**

I have carefully reviewed the rebuttals and the paper. While I acknowledge that the architecture design demonstrates a novel combination of existing components, the key question remains whether this offers new theoretical or practical insights to the community. Since there is no theoretical analysis in the paper, the practical impact becomes central to its value. Unfortunately, unresolved concerns regarding hyperparameter tuning, empirical performance, and presentation diminish the paper’s strength as a standalone practical contribution.

I believe this paper has potential but requires a major revision to address these issues. A carefully revised and resubmitted version could make a stronger contribution to the field.

**Additional Comments On Reviewer Discussion:**

Several concerns regarding the empirical evaluation, such as performance, hyperparameter tuning, and the validity of the problem setting, were raised by the reviewers. While the authors made significant efforts through their rebuttals and additional experiments, these concerns remain fundamentally unresolved. Considering my assessment noted above, I conclude that the paper requires at least one more round of major revisions. Therefore, I recommend rejecting the paper at this stage.

---

### Decision · Program_Chairs · 2025-01-22

Reject